



# Towards disentangling heterogeneous soil moisture patterns in Cosmic-Ray Neutron Sensor footprints

Daniel Rasche[1], Markus Köhli[3,4], Martin Schrön[5], Theresa Blume[1], and Andreas Güntner[1,2]

[1]GFZ German Research Centre for Geosciences, Section Hydrology, 14473, Potsdam, Germany
[2]University of Potsdam, Institute of Environmental Sciences and Geography, 14476, Potsdam, Germany
[3]Physikalisches Institut, Heidelberg University, Im Neuenheimer Feld 226, 69120 Heidelberg, Germany
[4]Physikalisches Institut, University of Bonn, Nussallee 12, 53115 Bonn, Germany
[5]UFZ – Helmholtz Centre for Environmental Research GmbH, Dep. Monitoring and Exploration Technologies, Permoserstr. 15, 04318, Leipzig, Germany

**Correspondence:** Daniel Rasche (daniel.rasche@gfz-potsdam.de)

**Abstract.**

Cosmic-Ray Neutron Sensing (CRNS) allows for non-invasive soil moisture measurements at the field scale. The derivation of soil moisture generally relies on secondary cosmic-ray neutrons in the epithermal-to-fast energy range. Most approaches and processing techniques for observed neutron intensities are based on the assumption of homogeneous site conditions within

the measurement footprint of the neutron detector.

In this study we investigated how a non-uniform soil moisture distribution within the footprint impacts the CRNS soil moisture estimation and how the combined use of epithermal and thermal neutrons can be advantageous in this case. Thermal neutrons have lower energies and a substantially smaller measurement footprint around the sensor than epithermal neutrons.

Analyses using URANOS neutron Monte-Carlo simulations to investigate measurement footprint dynamics at a study site

in north-eastern Germany revealed that the thermal footprint mainly covers mineral soils in the near-field to the sensor while the epithermal footprint also covers large areas with organic soils.

We found that either combining the observed thermal and epithermal neutron intensities by a rescaling method developed in this study, or adjusting all parameters of the transfer function leads to an improved calibration against reference soil moisture measurements in the near field compared to the standard approach and using epithermal neutrons alone. We also found that the

relationship between thermal and epithermal neutrons provided an indicator for footprint heterogeneity. We therefore suggest that the combined use of thermal and epithermal neutrons offers the potential of a spatial discretization of the measurement footprint in terms of near and far field soil moisture dynamics.



## 1 Introduction

Soil moisture is a key variable in the hydrological cycle (e.g., Vereecken et al., 2008, 2014; Seneviratne et al., 2010) driving i.e. energy fluxes, groundwater recharge, runoff generation processes and biomass production which in turn, influence climatic variables on varying spatio-temporal scales (e.g. see, Daly and Porporato, 2005; Vereecken et al., 2008; Seneviratne et al., 2010; Wang et al., 2018). Consequently, observations of soil moisture have a high importance for the estimation of landscape water balances and hydrological modelling. However, these applications would profit especially from field-scale observations

covering several hectares. At this scale the spatial (and temporal) resolution of satellite derived soil moisture products is too coarse and in-situ soil moisture sensors would need to be installed in very large numbers due to the high spatial variability of soil moisture (Famiglietti et al., 2008; Vereecken et al., 2014; Babaeian et al., 2019). In agricultural areas this in-situ installation is additionally hampered by management practices such as ploughing, tillage and harvest (Stevanato et al., 2019).

Introduced about a decade ago, Cosmic-Ray Neutron Sensing (CRNS) (e.g., Zreda et al., 2008; Desilets et al., 2010) partly

overcomes these issues and allows for non-invasive soil moisture estimation at the field scale. It provides a representative spatially averaged soil moisture value across the instrument's measurement footprint (Schrön et al., 2018b) of approximately 12 hectares. Resulting field-scale soil moisture products were successfully used for the calibration and validation of satellite derived soil moisture products, as well as improved land-surface and rainfall-runoff models (e.g., Holgate et al., 2016; Montzka et al., 2017; Iwema et al., 2017; Duygu and Akyürek, 2019; Dimitrova-Petrova et al., 2020). Combining soil moisture products

of different spatial scales may overcome scale gaps as exemplarily shown by Fersch et al. (2018). Roving CRNS-devices (e.g., McJannet et al., 2017; Schrön et al., 2018a; Vather et al., 2019) as well as dense sensor networks (e.g., Fersch et al., 2020; Heistermann et al., 2021) pose further opportunities for covering even larger areas.

Cosmic-Ray Neutron Sensing relies on the number of naturally occurring secondary cosmic-ray neutrons in the water-sensitive epithermal energy range from $> 0.2\,\mathrm{eV}$ to $1\,\mathrm{MeV}$ (Köhli et al., 2015) counted by a neutron detector above the soil

surface. Neutrons in the epithermal energy range are highly sensitive to the amount of hydrogen in the surrounding area due to their energy loss by elastic scattering processes. As a result, an increase in hydrogen results in a decrease of epithermal neutrons counted by the instrument as neutrons are slowed down more effectively. In turn, thermal neutrons have energies below $0.2\,\mathrm{eV}$ and show a more complex response to the dynamics of hydrogen and other elements. The interaction with hydrogen shows two competing effects: on the one hand, the thermal neutron abundance is positively correlated with the amount of hydrogen,

as more thermal neutrons are generated due to thermalization of epithermal neutrons. On the other hand, neutron absorption leads to a decrease of thermal neutrons with increasing hydrogen abundance (e.g., Hubert et al., 2016). As a consequence, thermal neutrons may show a similar response to variations in hydrogen and thus soil moisture (the largest terrestrial near-surface hydrogen storage). However, this response may be less distinct than the one of epithermal neutrons (Weimar et al., 2020) which are mainly driven by elastic scattering. An example for the more complex behaviour of thermal neutrons is the

moderation optimum describing the amount of hydrogen at which the thermalization is most effective (Hubert et al., 2016).

The measurement footprint size of CRNS varies with air pressure, air humidity and soil moisture conditions and ranges from 130 and 240 m radius with a depth of 15 to 83 cm during wet and dry conditions, respectively (Köhli et al., 2015). Additionally,





topographic features such as open water or strong topographic gradients may influence the footprint size (e.g., Köhli et al., 2015; Schattan et al., 2019; Mares et al., 2020).

Although neutrons in the epithermal energy range are the basis for deriving soil moisture contents, thermal neutrons remain in the focus of CRNS research as they can provide valuable information for estimating biomass (e.g., Tian et al., 2016; Jakobi et al., 2018; Vather et al., 2020) or snow water equivalent (SWE) (Bogena et al., 2020), e.g. by using the ratio of epithermal and thermal neutrons. Compared to epithermal neutrons, little is known about the behaviour of thermal neutrons. For instance, when thermal and epithermal neutrons are combined, a measurement footprint of similar size is assumed for both energy
ranges (e.g., Vather et al., 2020). However, the average footprint size of CRNS, e.g., the integration radius of thermal neutrons can be expected to be much smaller (approx. 35 m) compared to epithermal neutrons (200 m) (e.g. see, Bogena et al., 2020). Considering different footprint sizes of thermal and epithermal neutrons, a combination of both through calculating neutron ratios requires all hydrogen to be distributed homogeneously in the measurement footprints. As a result uncertainties may arise when hydrogen is not distributed homogeneously in the footprints as mentioned by Bogena et al. (2020) and this limits
the applicability of combining thermal and epithermal neutrons. This may be of particular importance as most studies with stationary CRNS assume homogeneous site conditions. For instance, neutron transport modelling so far assumes homogeneous soil water distributions when characterising footprint dynamics and weighting functions (e.g., Köhli et al., 2015; Schrön et al., 2017) or developing transfer functions for deriving soil moisture from epithermal neutron intensities (e.g., Desilets et al., 2010; Franz et al., 2013; Andreasen et al., 2020; Köhli et al., 2021). However, different footprint sizes may offer the opportunity
of a horizontal differentiation between near and far-field soil moisture dynamics. Although previous studies confirmed the applicability of CRNS at heterogeneous study sites for deriving spatially averaged soil moisture time series (e.g., Franz et al., 2016; Sigouin et al., 2016; Schrön et al., 2017; Pang et al., 2021), approaches for the spatial disaggregation of CRNS-derived soil moisture values at heterogeneous observation sites have not been assessed in detail yet.

Against this background, this study investigates the footprint size and neutron dynamics of epithermal and thermal energies
at a heterogeneous study site in the TERENO Lowland Observatory in north-eastern Germany. Consisting of mineral soils in the near-field and partly surrounded by groundwater-influenced organic peatland soils in the far-field, different approaches for a spatial discretization of the measurement footprint can be tested at this site. This is aided by the distinct hydraulic characteristics of organic peatland soils (e.g., Dettmann et al., 2014; Rezanezhad et al., 2016) and mineral soils which lead to different soil water dynamics and water contents.

Due to the general decrease of thermal neutron count rates with increasing soil moisture but with smaller integration radius, we hypothesize that Spearman's rank correlation coefficient between normalized thermal and epithermal neutron intensities can serve as measure for footprint heterogeneity. Secondly, we hypothesize that both, adjusting the neutron transfer function to near-field soil moisture observations or a combination of the normalized epithermal and thermal neutron intensities allow for a spatial discretization of the measurement footprint. To test these hypotheses, we first set up Monte-Carlo neutron transport
simulations using the URANOS code (Köhli et al., 2015). This code is often used in CRNS research (e.g., Köhli et al., 2015, 2021; Schrön et al., 2017, 2018a, b; Schattan et al., 2019; Li et al., 2019; Weimar et al., 2020) to develop transfer functions and weighting procedures. In our study, we use it to identify the footprint size and dynamics of neutrons in the thermal and





epithermal energy ranges under different soil moisture conditions at the heterogeneous study site. Secondly, we adjust the standard transfer function used for deriving soil moisture from neutron observations and apply a combination of observed

thermal and epithermal neutrons in order to improve the calibration of CRNS-derived soil moisture estimates against reference soil moisture observations in the near-field. Finally, we illustrate the potential of deriving differentiated soil moisture dynamics under heterogeneous footprint conditions by either adjusting the transfer function or by adjusting the neutron signal.

## 2 Material and methods

### 2.1 Study site

The study site is located in the Terrestrial Environmental Observatory TERENO-NE (Zacharias et al., 2011; Heinrich et al., 2018) in the lowlands of north-eastern Germany (Fig. 1). The average annual temperature is 8.8 °C and rainfall amounts to 591 mm per year at Waren weather station approx. 35 km away from the study site (station ID: 5349, period 1981–2010) (DWD - German Weather Service, 2020a, b). Geologically, the study site is situated on a glacial outwash plain south of a terminal moraine formed during the Pomeranian phase of the Weichselian glaciation (Börner, 2015). Within the outwash plain we still

find non-eroded outcrops of glacial till of previous glaciation phases while fens formed subsequently in depressions and local sinks (Börner, 2015) due to rising temperatures and groundwater levels in the Holocene.

The CRNS site (Site A) itself is located on a slightly elevated outcrop of Weichselian glacial till surrounded by peatland (Fig. 1). Dominating soil types in areas with glacial till are cambisols formed on sandy loam while the peatland areas are characterized by histosols rich in clay and silt and low water table depths. Soil samples were taken from mineral and organic

soils in depths from 0–30 cm in 5 cm increments at 21 random locations within a 200 m radius in February 2020 and were analysed for retrieving local soil properties. Within 10 m, 10 to 50 m and 50 to 200 m radius 5, 6, and 10 samples were taken, respectively, thus matching the decreasing sensitivity of the neutron detector with increasing radius. The analyses revealed an average bulk density of 1.43 g cm$^{-3}$ in areas with mineral and 0.29 g cm$^{-3}$ in areas with organic soils. The site average bulk density calculated from all available samples is 1.11 g cm$^{-3}$. Based on the material density of quartz, these values were used

to estimate soil porosities of 89 and 46 percent for organic and mineral soils. The average percentage of soil organic carbon determined from loss-on-ignition analyses (550 °C, 24 hours) revealed 0.70 g g$^{-1}$ for organic soils and 0.02 g g$^{-1}$ for mineral soils. Based on these soil samples, the average gravimetric water content in organic soils in the far-field was 0.62 m$^3$ m$^{-3}$ while it was only 0.15 m$^3$ m$^{-3}$ for near-field mineral soils illustrating the two distinct soil moisture regimes at the study site. Regardless of soil type, pasture is the prevailing type of land-cover mainly used for cattle grazing while larger areas forested

with *Pinus sylvestris* are found in greater distances of more than 1 km towards the east (Fig. 1). The observation site is one of three sites in the TERENO-NE observatory equipped with a CRS1000 neutron detector (Hydroinnova LLC, USA) and also includes a weather station that permanently monitors relative humidity, wind speed, temperature as well as long and short-wave solar radiation. Additionally, irregular monthly groundwater measurements are available. The other two CRNS observation sites (B and C, Fig. 1b) represent forest sites with a rather homogeneous distribution of soil moisture in the measurement footprint

of the CRNS. Further details regarding the site C can be found in Heidbüchel et al. (2016).





**Figure 1.** Location of the study area and within Germany (a), positions of all CRNS locations of the TERENO lowland observatory in the Mueritz National Park (b) and the heterogeneous CRNS observation site of this study (c) (digital elevation model: LAIV-MV - State Agency for Interior Administration Mecklenburg-Western Pomerania (2011), land cover: BKG - German Federal Agency for Cartography and Geodesy (2018)).

The neutron detectors are composed of two proportional counter tubes filled with $^3$He gas (see Zreda et al. (2012) and Schrön et al. (2018b) for a detailed description). One bare, unshielded tube to detect neutrons in the thermal energy range and a second, moderated counter tube shielded with a 2.5 cm high-density polyethylene housing to measure neutrons in the epithermal energy range. It should be noted, that the bare counter tube may observes about 5 percent epithermal neutrons and the moderated tube observes up to 45 percent thermal neutrons (Andreasen et al., 2016) which has to be considered when comparing different detector systems and the results of neutron transport simulations. Time series of in-situ soil moisture point sensors of two different types are available at the site. A total of 6 SMT100 sensors (Truebner GmbH, Germany) are installed with 2 sensors each in 10, 20, and 30 cm depth. They record soil moisture in 10 min intervals. TDR sensors (Campbell Scientific Ltd., UK) are






installed in the same depths, with 4 sensors in 10 cm depth, 3 sensors in 20 cm and 5 sensors in 30 cm depth. The record interval

for the TDR probes is 15 min. Measurements are converted to soil moisture using the manufacturers' calibrations. All point

sensors are installed within 30 m radius around the neutron detector and thus only cover the near-field composed of mineral

soils. Continuous reference observations from the far-field peatland soils are not available. Given the higher noise level of the

TDR time series but soil moisture dynamics that are very similar to the SMT100 sensors in the respective soil depths, only

SMT100 data are used in the following analyses and presented in the manuscript. The identical processing procedure was used

for the soil moisture time series from TDR soil moisture sensors (for results see appendix).

## 2.2 Neutron simulations

In the present study, we apply the Monte-Carlo based neutron transport model URANOS (version v0.9 $\omega$18, see Köhli et al.

(2015) for details). By simulating 200 million source neutrons we intend to estimate the influence of water content variations

in areas with organic soils in the far-field of the neutron detector on the neutron flux and footprint size of epithermal and

thermal neutrons. The model set-up uses a simplified representation of soil distributions (Fig. 1) within a rectangular 900 by

900 m sized model domain with a horizontal resolution of 1 m around the neutron detector. Three simplifications had to be

made in order to set up the model: a flat topography was assumed, soil porosities derived from field samples were assumed

to be valid for the entire simulated soil column of 2 m depth and organic soils are only differentiated by their significantly

higher porosities while their chemical composition equals that of mineral soils. This last simplification is due to limitations

of the neutron transport model. Additionally, all simulations were made with a single set of atmospheric boundary conditions,

namely an assumed cutoff rigidity of 3 GeV based on Andreasen et al. (2017a), an absolute humidity of 8.3 g m$^{-3}$ and an

atmospheric shielding depth of 1028.5 g cm$^{-2}$. In case of absolute humidity and atmospheric shielding depth, these values

represent site averages derived from local measurements for the study period from 2015 to 2018.

On the one hand, the simulation scenarios described in the following sections allow for investigating thermal and epithermal

measurement footprint changes when soil moisture is either stable or dynamic in the near-field where the CRNS method is

most sensitive. Additionally, we can derive potentially valuable information on neutron intensity variations when the water

content varies at different rates in the near-field and far-field. This may be of particular importance at study sites influenced by

peatland soils as these are characterised by i.e. higher storage capacities. An overview of all simulation sets and the included

simulation scenarios performed in the scope of this study can be found in Table 1.

### 2.2.1 Simulation set 1: Static near-field soil moisture, variable far-field soil moisture

In the first simulation set that consists of 7 neutron transport simulation scenarios, the simulated soil moisture was kept constant

in the near-field mineral soil areas. The soil moisture in areas with peatland soils was altered for each scenario and ranges from

0.1 to 0.7 m$^3$ m$^{-3}$ across the entire soil column at a porosity of 89 percent. During all scenarios, the soil moisture in areas with

mineral soil remains constant at 0.1 m$^3$ m$^{-3}$ at a porosity of 46 percent (Table 1). This fixed low water content in the near-field

was chosen as the measurement radius from which detected thermal or epithermal neutrons originate can be expected to be

largest at dry soil conditions (Köhli et al., 2015; Schrön et al., 2017). Therefore, the largest influence of far-field soil water





dynamics on neutron count rates at the detector location can be expected at low soil moisture conditions in the near-field. Hence, simulating constant low near-field soil water contents and solely varying far-field soil water contents allows for the isolated investigation of the impact of peatland soil water variations on the observed neutron intensities, as well as the corresponding

footprint variations.

To investigate the footprint variability caused by soil water changes in peatland soils of the far-field, we calculate the footprint radius as the 86 percent quantile of distances ($R_{86}$) to the detected neutron origins for thermal (0.001 eV to 0.2 eV) and epithermal (> 0.2 eV to 0.01 MeV) neutrons. For detected epithermal neutrons, the distance to the point of first soil contact is considered as the point of origin as secondary epithermal neutrons generated from nuclear evaporation processes in the soil

are sensitive to hydrogen by elastic scattering (e.g. Köhli et al., 2015). In contrast, to our knowledge, the definition of the origin of detected thermal neutrons has not been assessed in detail yet. On the one hand, the point of thermalization (i.e. point where a neutron first reached an energy in the thermal range) may be a suitable definition of the origin because as neutrons reach thermal energies, absorption adds to elastic scattering as a second important interaction process between neutrons and hydrogen as well as matter in general. On the other hand, if thermal neutrons are generated from higher energetic epithermal

neutrons, the amount of detected thermal neutrons may partly be influenced by the amount of epithermal neutrons and their origin. As a consequence, we apply and compare both options, the point of thermalization and the point of first soil contact, for defining the origin of detected thermal neutrons and the resulting footprint radius. The measurement depth of epithermal and thermal neutrons is derived similarly. For thermal neutrons, the measurement depth $D_{86}$ is defined as the 86 percent quantile of either the depth of the thermalization point or the maximum depth along the neutron transport path while for epithermal

neutrons we use only the latter.

It should be noted that the virtual detector in the simulations is significantly larger (9 m radius) than a real neutron detector in order to enhance the count rate and decrease the computational time. However, this means that neutrons originating below the detector (i.e. originating within the 9 m radius) are considered to have a travel distance of zero meters.

### 2.2.2 Simulation sets 2 and 3: Varying soil moisture in both the near-field and far-field

The second set of simulation scenarios included variations of soil moisture contents in the far field as well as in the near-field. All other simulation parameters remained equal to previous simulations. We simulate 6 simulation scenarios with an equal decrease of soil water contents in near-field mineral soils and far-field peatland soils. The highest soil moisture content simulated is 0.35 m$^3$ m$^{-3}$ for mineral soils and and 0.70 m$^3$ m$^{-3}$ for peatland soils. These soil water contents are decreased in equal intervals of 0.05 m$^3$ m$^{-3}$ in both soils (Table 1). In the third set of simulations we investigate the effect when soil

moisture is reduced more strongly in peatland soils compared to mineral soils. Here, the soil moisture in peatland areas is reduced in 0.10 m$^3$ m$^{-3}$ intervals from 0.70 to 0.20 m$^3$ m$^{-3}$ while soil moisture in the mineral soils is reduced from 0.35 to 0.10 m$^3$ m$^{-3}$ in 0.05 m$^3$ m$^{-3}$ intervals.





**Table 1.** Overview of the different Monte-Carlo based neutron transport simulation scenarios conducted within the different simulation sets. Near field soil moisture refers to soil moisture in areas covered with mineral soils while far field soil moisture refers that in the peatland soils.

| Simulation set | Simulation scenario no. | Near field soil moisture [$m^3\ m^{-3}$] | Far field soil moisture [$m^3\ m^{-3}$] |
|---|---|---|---|
| | 1 | 0.10 | 0.70 |
| | 2 | 0.10 | 0.60 |
| 1: Static | 3 | 0.10 | 0.50 |
| near field | 4 | 0.10 | 0.40 |
| soil mositure | 5 | 0.10 | 0.30 |
| | 6 | 0.10 | 0.20 |
| | 7 | 0.10 | 0.10 |
| | 1 | 0.35 | 0.70 |
| 2: Equal | 2 | 0.30 | 0.65 |
| decrease in near | 3 | 0.25 | 0.60 |
| and far field | 4 | 0.20 | 0.55 |
| | 5 | 0.15 | 0.50 |
| | 6 | 0.10 | 0.45 |
| | 1 | 0.35 | 0.70 |
| 3: Unequal | 2 | 0.30 | 0.60 |
| decrease in near | 3 | 0.25 | 0.50 |
| and far field | 4 | 0.20 | 0.40 |
| | 5 | 0.15 | 0.30 |
| | 6 | 0.10 | 0.20 |

## 2.3 In-situ neutron observations

### 2.3.1 Processing of measured neutron intensities

For our 4-year study period from January 2015 to December 2018 neutron intensities were aggregated from sub-hourly intervals to hourly values and smoothed by a 13-hour moving average in order to reduce noise in the data (e.g., Bogena et al., 2013). Data gaps were caused by power cuts, technical issues or maintenance activities. Outliers were identified by a threshold of four times the standard deviation and were excluded from the analyses. Raw neutron observations were corrected for variations in atmospheric shielding depth or air pressure and primary neutron influx (e.g. Zreda et al., 2012). The standard correction

procedure for air humidity is defined for neutrons in the epithermal energy range (Rosolem et al., 2013) and may not be valid for thermal neutrons. As a consequence, in this study, we corrected thermal and epithermal neutron intensities only for variations in atmospheric shielding depth and primary neutron influx in order to maintain comparability. It should be noted that the correction procedures applied to thermal and epithermal neutron intensities differ among previous studies (e.g. Andreasen et al., 2016;

Jakobi et al., 2018) and illustrate the need for further research. To correct raw neutron intensities for varying atmospheric

shielding depths, air pressure values in hPa measured by the neutron detector are converted to atmospheric shielding depth in

$\mathrm{g\,cm^{-2}}$ by multiplication with $1.0194\,\mathrm{s^2\,m^{-1}}$ (Heidbüchel et al., 2016). The required reference value is the average atmospheric

shielding depth for the 4-year study period and the attenuation length ($135.6\,\mathrm{g\,cm^{-2}}$) is adapted from Heidbüchel et al. (2016).

The correction for variations in primary neutron flux is done using pressure and efficiency corrected primary neutron data from

the Jungfraujoch neutron monitor in Switzerland (JUNG, www.nmdb.eu). Again, the reference value is defined as the average

influx during the study period. The corrected thermal (bare) and epithermal (moderated) neutron intensities for the study period

are illustrated in Fig. 2.

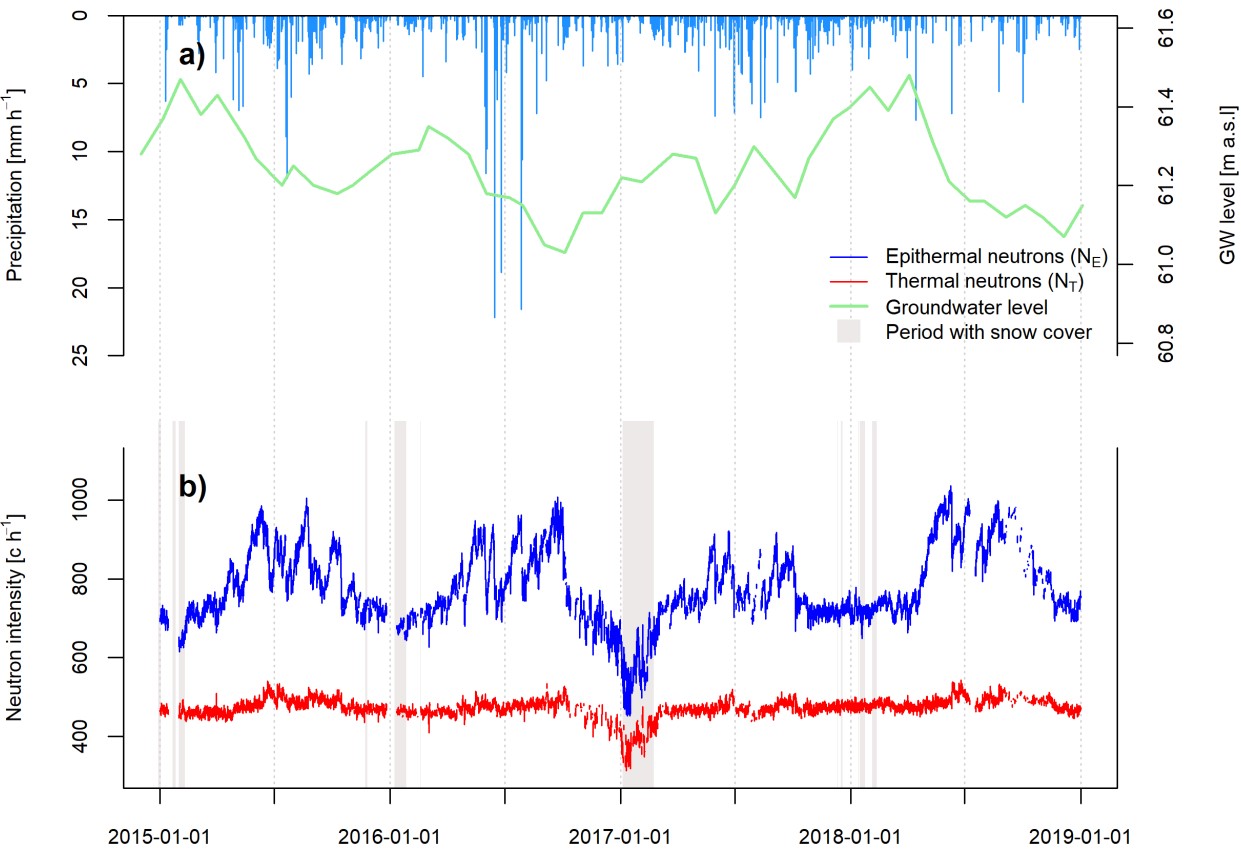

**Figure 2.** Field observations for the period from January 2015 to December 2018: a) hourly rainfall and monthly groundwater level measurements, b) the corrected neutron intensities in the thermal ($N_T$) and epithermal energy range ($N_E$) (b).





Corrected epithermal neutron intensities can be converted to volumetric soil moisture $\theta_{\mathrm{CRNS}}$ in $\mathrm{m^3\ m^{-3}}$ using the standard transfer function introduced by Desilets et al. (2010):

$$\theta_{\mathrm{CRNS}} = \left( \frac{a_0}{\frac{N}{N_0} - a_1} - a_2 \right) \times \frac{\rho_{\mathrm{soil}}}{\rho_{\mathrm{water}}}. \tag{1}$$


Here, $a_0$ (0.0808), $a_1$ (0.372) and $a_2$ (0.115) are the shape-defining parameters of the hyperbolic transfer function. $N$ describes the corrected neutron intensity, $\rho_{\mathrm{soil}}$ is the average soil bulk density in the measurement footprint ($\mathrm{kg\ m^{-3}}$), $\rho_{\mathrm{water}}$ is the density of water in $\mathrm{kg\ m^{-3}}$ and $N_0$ is a free calibration parameter describing the site-specific neutron intensity over dry soil at $0.0\,\mathrm{m^3\ m^{-3}}$. In this study, we used the revised standard transfer function recently introduced by Köhli et al. (2021) which

provides a physical meaning for each of the shape-defining parameters from eq. (1). The three variables in the revised transfer function (eq. (2)) can be calculated from the variables of the standard transfer function.

$$\theta_{\mathrm{CRNS}} = \left( \tilde{a_0} \frac{1 - \frac{N}{N_{\mathrm{max}}}}{\tilde{a_1} - \frac{N}{N_{\mathrm{max}}}} \right) \times \frac{\rho_{\mathrm{soil}}}{\rho_{\mathrm{water}}}, \tag{2}$$

where

$\quad \tilde{a_0} = -a_2, \tag{3}$

$$\tilde{a_1} = \frac{a_1 a_2}{a_0 + a_1 a_2}, \tag{4}$$

$$N_{\mathrm{max}} = N_0 \times \frac{a_0 + a_1 a_2}{a_2}. \tag{5}$$

The calibration procedure is performed for the entire time series of observed, corrected neutron intensities of epithermal neutrons $N_E$. Two options for deriving soil moisture based on the standard transfer function (eq. (1)) are possible. The first option

requires eq. (1) to be solved for $N_0$ in order to approximate the calibration parameter. The calculated $N_0$ is then used in eq. (1). In the second procedure, the $N_0$ parameter of eq. (1) is calibrated iteratively against reference soil moisture measurements in order to derive the site-specific neutron intensity over dry soils $N_0$.

We used the revised transfer function (eq. (2) – (5)) and iteratively calibrated $N_0$ because continuous in-situ soil moisture measurements offer a high number of reference points. Regardless of which equation is used for deriving soil moisture values

from neutron observations and corresponding calibration option, reference soil moisture observations need to be weighted





according to their depth and distance to the neutron detector in order to match its sensitivity regarding the origin of epithermal neutrons (Schrön et al., 2017).

We weighed all available point sensor measurements per time step in order to derive a depth-distance weighted soil moisture time series after Schrön et al. (2017). The weighting approach takes e.g. soil moisture, bulk density (1.43 g cm$^{-3}$), additional
hydrogen pools (organic carbon of 0.02 g g$^{-1}$ and lattice water of 0.001 g g$^{-1}$), air humidity and vegetation height (0.2 m) into account. Hourly time series of all SMT100 soil moisture sensors in depths 10, 20 and 30 cm for all snow-free periods from January 2015 to December 2018 were weighted accordingly. For SMT100 probes, we also excluded soil moisture observations during soil temperatures below zero degrees Celsius. Lastly, the $N_0$ is iteratively adjusted to derive a soil moisture time series from observed neutron intensities resulting in the highest goodness-of-fit in terms of the highest Kling-Gupta-Efficiency
(KGE) (Gupta et al., 2009) compared to the weighted reference time series from the in-situ sensors in the near-field. To assess the impact of the weighting on the calibration result we also compare the depth-distance weighted calibration with a calibration based on the arithmetic mean of all available in-situ sensors. This calibration approach is referred to as the standard calibration approach throughout the manuscript.

### 2.3.2 Improving the CRNS-derived soil moisture estimation

To achieve a better calibration result against the observations of the reference sensors in the near-field, we adjusted the shape of the transfer function by tuning the parameters $a_0$, $a_1$ and $a_2$. Tuning all shape-defining parameters was done in previous studies and resulted in a better goodness-of-fit between CRNS-derived soil moisture values and reference measurements (e.g., Rivera Villarreyes et al., 2011; Lv et al., 2014; Heidbüchel et al., 2016; Tan et al., 2020). In this study, we adjust parameters $N_0$, $a_0$, $a_1$ and $a_2$ using a Monte-Carlo based approach by testing 10,000 quasi-random combinations of the parameters and selecting
the parameter set producing the highest statistical goodness-of-fit in terms of the KGE. This approach will be referred to as alternative approach 1 in the following sections.

Furthermore, we tested a second approach where we made use of the simultaneously recorded and corrected thermal neutron intensity $N_T$ to create a rescaled neutron time series $N_{ET}$ using equation (6):

$$N_{ET} = \left( \frac{N_E + N_T}{N_E + N_T} \right) \times \overline{N_E} \tag{6}$$


Here, $\overline{N_E + N_T}$ is the average of the sum of epithermal and thermal neutrons while $\overline{N_E}$ is the average of the epithermal neutron intensity only. The result from equation (6) is a neutron time series which averages the dynamics of thermal and epithermal neutron intensities and thus leads to a rescaled epithermal neutron time series $N_{ET}$ which now shows a different relationship with the reference soil moisture measurements characterised by a shallower slope compared to $N_E$. The calibration
was performed iteratively by adjusting $N_0$ only. In terms of normalized neutron intensity dynamics, this equation is equal to summing the absolute epithermal and thermal neutron intensity which would consequently lead to a much higher $N_0$ after calibration. The latter approach will be referred to as alternative approach 2 for the remainder of the manuscript.





To test if the CRNS-derived soil moisture from one of the alternative approaches 1 and 2 differs significantly from the traditional standard approach using $N_E$ and calibrating $N_0$ only, we performed a time series comparison based on bootstrapping
residuals and the Wilcoxon rank sum test. First, the CRNS-derived soil moisture time series based on (i) $N_E$ and calibrating $N_0$, (ii) $N_E$ and calibrating all parameters as well as (iii) $N_{ET}$ and calibrating $N_0$ were smoothed with a normal Nadaraya-Watson kernel regression smoother using a large bandwidth of 1,000 in order to achieve an intense smoothing effect. Missing values were excluded before applying the smoothing algorithm and reintroduced to the smoothed time series. Then the residuals were calculated between the smoothed and original soil moisture time series. Next, a random sample of 5,000 residuals was
generated and used to produce a quasi-random distribution of 5,000 soil moisture values per time step for each of the smoothed time series of the three variants mentioned above. This is done to receive a distribution of soil moisture values per time step of each CRNS-derived soil moisture time series which can be compared in the next step. For each time step, an unpaired Wilcoxon rank sum test was performed in order to determine time steps where significant differences ($p < 0.05$) occur between the CRNS-derived soil moisture time series calculated with the classic approach and with the two alternative approaches.

In a first attempt to find a measure to characterise footprint heterogeneity, the Spearman's rank correlation coefficients between normalized thermal and epithermal neutron counts were calculated for the study site and for the two nearby CRNS sites that are located in forested terrain (see Fig. 1) with rather homogeneous soil characteristics in their footprint. Neutron intensites of those two sites were corrected in the same way as described above. Unless otherwise stated, all calculations were performed in R statistical software (R Core Team, 2018) using for instance the hydroGOF package (Zambrano-Bigiarini, 2017)
for calculating goodness-of-fit parameters.

## 3 Results

### 3.1 Simulated neutron response to soil moisture changes in the far-field peatland soils

The results of all neutron transport simulation scenarios with a constant soil moisture in mineral soils of the near-field reveal that the sensitive measurement footprint radius is distinctively smaller for thermal neutrons than for epithermal neutrons (Fig. 3).
The footprint radius of epithermal neutrons decreases from 153 m in the 0.1 m$^3$ m$^{-3}$ scenario to 123 m in the 0.7 m$^3$ m$^{-3}$ scenario with an average $R_{86}$ of 134 m based on all simulation scenarios. In comparison, the thermal footprint radius is distinctively smaller but depends on the definition of a detected neutrons origin in the model domain. If the point of thermalization is considered as the point of origin, the thermal footprint only exhibits a minor change in simulation scenarios with increasing soil moisture in the peatland soils of the far-field. It remains rather constant with a minimum and maximum $R_{86}$ of 49 m and
50 m, respectively (Fig. 3). The average footprint radius of thermal neutrons from all simulation scenarios is 49 m and thus, the average epithermal footprint radius is 2.7 times larger than the thermal footprint radius. However, if the first soil contact is considered as the point origin, the thermal footprint is larger and decreases with increasing peatland soil moisture from 114 m and 88 m with an average footprint radius of 96 m. Thus, the neutron transport simulations led to an average horizontal integration area of 0.8 ha and 2.9 ha for thermal neutrons depending on which footprint definition is applied. The average
horizontal integration area of epithermal neutrons has a size of 5.6 ha. The average integration depth $D_{86}$ remains constant for





thermal neutrons with an average $D_{86}$ of 0.27 m if the point of thermalization is considered as the origin while it is distinctively larger if the maximum depth is used. Here, the average measurement depth increases to 0.52 m and is larger than the average measurement depth of epithermal neutrons revealing a $D_{86}$ of 0.3 m. Even though thermal neutrons show a smaller integration radius, the integration depth might be larger than that of epithermal neutrons.

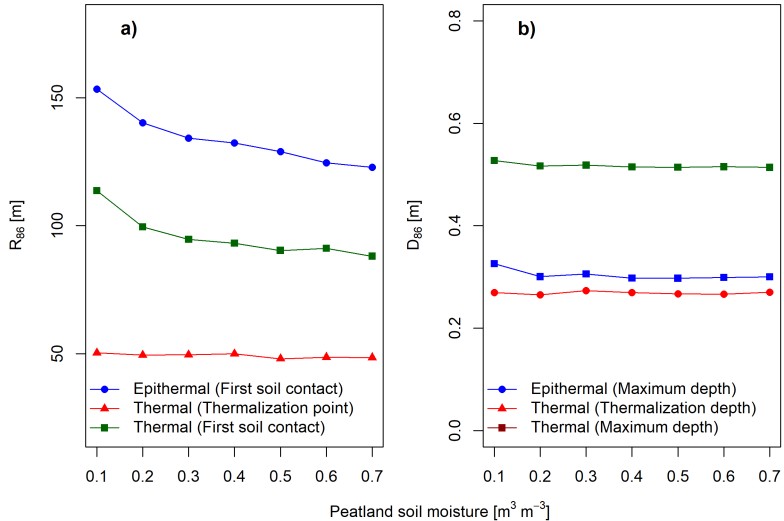

**Figure 3.** Simulation results for the measurement footprint radius (a) and depth (b) of detected thermal and epithermal neutrons.

The number of neutrons detected by the virtual detector decreases with increasing soil moisture in the peatland soils (Fig. 4) during the different simulation scenarios from 0.1 to 0.7 m³ m⁻³. The total number of detected neutrons, i.e. of neutrons with any origin in the model domain, illustrates the decrease of detected epithermal and thermal neutrons with increasing peatland soil moisture and a generally lower number of detected thermal neutrons (Fig. 4). The number of detected epithermal neutrons decreases by 9.5 percent from about 21800 to 19700 in the 0.1 and 0.7 m³ m⁻³ scenario. The amount of thermal neutrons
detected decreases to a lesser degree by 5.4 percent from about 17400 to 16500. Thus, the total number of epithermal neutron decreases 1.8 times stronger than the number of thermal neutrons with increasing far field soil moisture.

The visible influence of far field soil moisture variations on both epithermal and thermal neutrons raises the question of the impact on the fraction of detected neutrons, i.e. the fraction of detected neutrons originating from areas covered with peatland soils and minerals soils. We investigated the influence of peatland soil moisture variations in the far-field on the fractional
contribution to the total number of thermal neutrons when either the point of thermalization or the point of first soil contact is considered as the origin of the specific neutron. The latter represents the position of the first soil contact of a neutron within its life cycle in the model domain, i.e., the position where simulated neutron had its first soil contact before it further slowed down and reached the virtual detector as a neutron with thermal energy. For detected epithermal neutrons, the fraction of neutrons with peatland origin decreases with increasing peatland soil moisture from 19 to 12.5 percent and contributes on average 14.7





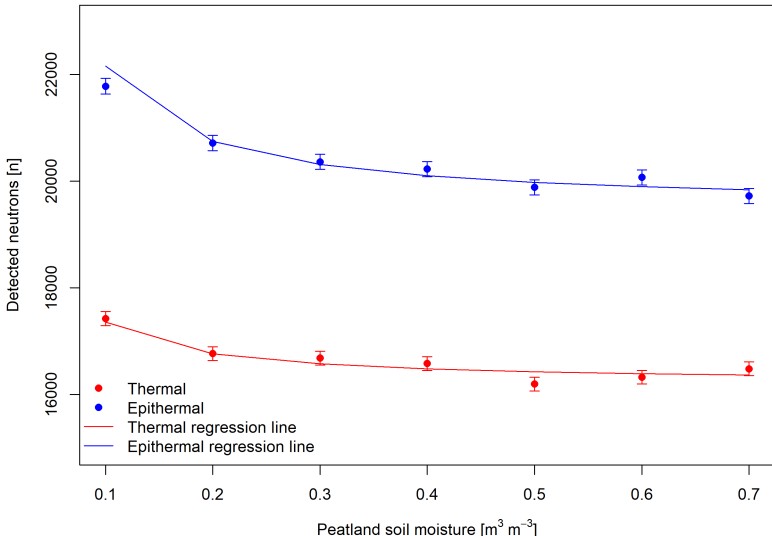

**Figure 4.** Total number of neutrons in the thermal and epithermal energy range observed by the virtual detector per simulated peatland soil moisture

percent to the total number detected epithermal neutrons (Fig. 5). For thermal neutrons the fraction of neutrons originating from peatland areas depends on the definition of the point of origin. If the point of thermalization is used, the average fraction of thermal neutrons originating from peatland areas is much lower with only 4.2 percent. Furthermore, the fraction does not change with increasing peatland soil moisture. In contrast, if the point of first soil contact is used for thermal neutrons as well, the contribution of thermal neutrons with peatland origin decreases from 13.6 to 8 percent with increasing soil moisture. On
average, the contribution from peatland is 9.7 percent.

### 3.2    Simulated neutron response to soil moisture changes in both near-field mineral and far-field peatland soils

In addition to keeping soil moisture values in mineral soils of the near-field constant, we simulated a second set of scenarios where both, near-field and far-field soil water contents were adjusted (Table 1). The results of the measurement radius and depth for the same decrease of soil moisture in areas covered with mineral soils and peatland soils can be found in Fig. 6. The
epithermal $R_{86}$ does not show a visible change if soil moisture is reduced equally and reveals an average footprint radius of 130 m. An increase of the $R_{86}$ can be observed when soil moisture is decreased twice as much in the peatland soils compared to the decrease in the mineral soils. In this case, $R_{86}$ increases from 131 to 140 m. A similar behaviour can be observed for thermal neutrons if the point of first soil contact is considered as the origin in the model domain. While in this case the $R_{86}$ remains constant at 90 m if soil moisture is decreased equally in mineral and peatland soils, $R_{86}$ increases from 88 to 100 m when soil
moisture is reduced twice as much in peatland compared to mineral soil areas. Similarly to the simulation set described in the previous chapter where the soil moisture is kept constant in the near field, the thermal $R_{86}$ is much smaller and does not change





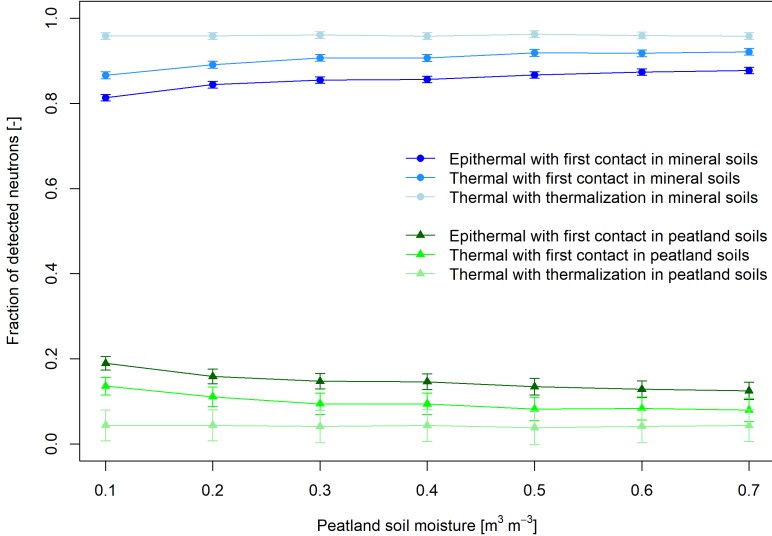

**Figure 5.** Fraction of detected epithermal and thermal neutrons with increasing soil moisture originating from areas covered with peatland soils and mineral soils in the model domain. For epithermal neutrons, the point of origin is defined as the point of first soil contact while for thermal neutrons both calculations, for the point of first contact and the point of thermalization are shown.

with varying soil moisture when the point of thermalization is considered. It remains constant at an average value of 49 m for the simulation sets where soil moisture is decreased both equally and unequally.

Unlike for the previous scenarios with constant near field soil moisture, the measurement depth varies noticeably in both

neutron energy ranges when soil moisture is also reduced in the near-field of the neutron detector. The epithermal integration depth $D_{86}$ increases from 0.11 to 0.30 m while the thermal integration depth changes from 0.11 to 0.27 m if the depth of the point of thermalization is used for defining the measurement depth. In contrast, if the thermal integration depth is calculated like it is done for epithermal neutrons, using the maximum depth along the neutron transport path, the thermal integration depth becomes much larger and exceeds the integration depth of epithermal neutrons. It increases from 0.21 to 0.51 if soil moisture

is reduced equally and to 0.52 m if soil moisture is reduced unequally in near-field mineral and far-field peatland soils.

A stronger increase of the detected epithermal neutrons normalized by the average detected neutrons of all simulation scenarios of set 2 and 3, respectively, can be observed when soil water contents decreases twice as fast in peatland soils compared to mineral soils (Fig. 7). Simulated thermal neutrons exhibit a more complex behaviour: the number of thermal neutrons increases with decreasing soil moisture in the model domain when the general soil water content is high, however,

when the overall soil moisture in the model domain is low, the detected thermal neutrons tend to either remain constant or even decrease if soil moisture is decreased stronger in peatland soils. Overall, the thermal neutrons tend to increase with decreasing soil water content.

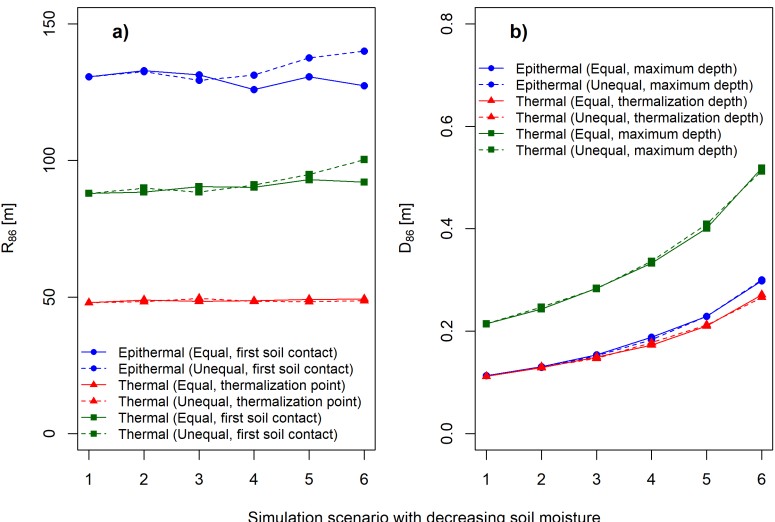

**Figure 6.** Simulated measurement footprint radius $R_{86}$ (a) and depth $D_{86}$ (b) of thermal and epithermal neutrons when soil moisture in areas with mineral and peatland soils decreases by the same amount (solid lines), and when peatland soil moisture decreases twice as much (dashed lines).

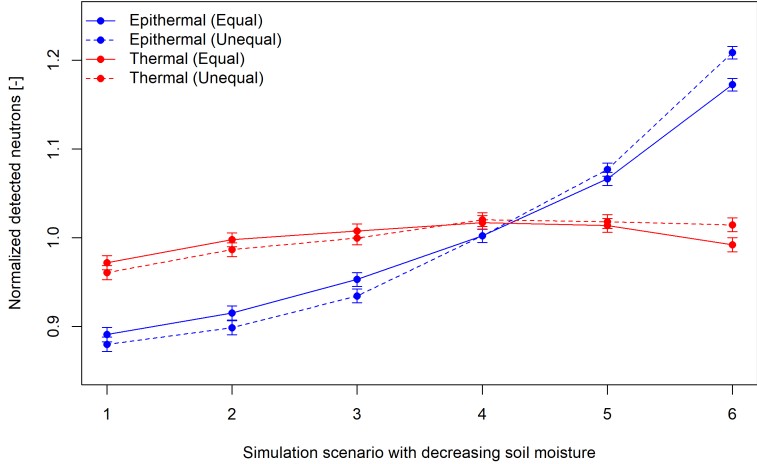

**Figure 7.** Simulated normalized thermal and epithermal neutron response when soil moisture in areas covered with mineral and peatland soils decreases in equal intervals (solid lines) and when peatland soil moisture decreases twice as much (dashed lines).





### 3.3 Relationship between thermal and epithermal neutron observations

Spearman's rank correlation coefficient is calculated between the normalized corrected hourly intensities of thermal and ep-
ithermal neutrons for the observation site (Site A) (Fig. 8a) and for comparison also at the two other nearby CRNS observation
sites (sites B and C) (Fig. 8b,c). The Spearman rank correlation coefficient for the other two sites is 0.95 showing a high
correlation between the neutrons observed by the shielded (epithermal) and unshielded (thermal) counter tube. In contrast, the
correlation coefficient at our main observation site is much lower, with only 0.58 (Fig. 8). Figure 8 illustrates the relation-
ship between relative observed neutron intensities in both energy ranges per study site. Apart from higher Spearman's rank
correlation coefficients, the point clouds for both sites that are assumed to have more uniform soil moisture (Fig. 8b,c) are
close to the 1:1 line although a slight non-linearity is visible. In contrast, the scatter plot for site A (Fig. 8a) reveals a strong
heteroscedasticity with deviations occurring during high relative neutron count rates in the epithermal and thermal energy
range.

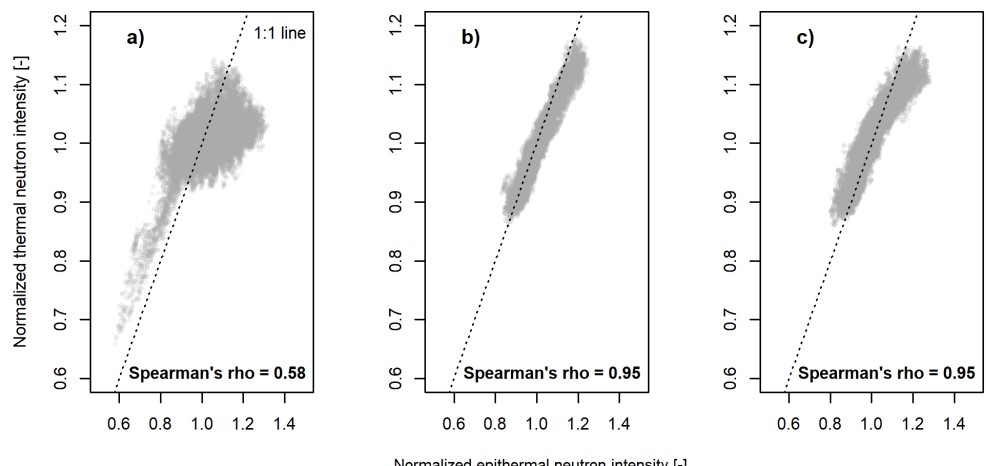

**Figure 8.** Relationship between observed normalized thermal and epithermal neutron intensity at the three CRNS sites: a) site A, the main
study site, b) site B (soil moisture assumed to be uniform) , c) site C (soil moisture assumed to be uniform). Normalized intensities were
calculated by dividing by the respective time series mean.

Adding the simulated normalized detected neutrons to the scatter plot of site A results in all simulated data points being
located within the range of observed values (Fig. 9). Data points of simulation sets 2 and 3 with varying near field soil moisture
cross the 1:1 line while the data points of simulation set 1, where the near-field soil moisture was kept constantly low, remains
on the right side of the 1:1 line.

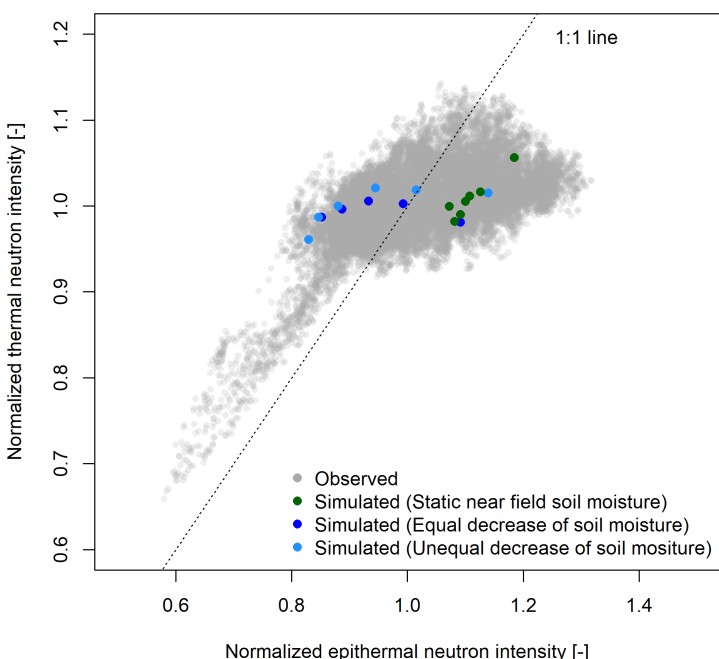

**Figure 9.** Relationship between normalized thermal and epithermal neutron intensities for in-situ observations and relationship between the normalized detected epithermal and thermal neutrons for simulated data. The simulated values refer to the simulation set and scenarios summarised in Table 1. Simulated neutrons are normalized by the average number of detected neutrons of all simulations in the respective energy range.

### 3.4 Estimation of soil moisture from observed neutron intensities

We estimated soil moisture time series from the CRNS signal with three different calibration approaches. We first applied the standard calibration approach by using the corrected epithermal neutron intensities ($N_E$) and iteratively calibrating $N_0$ in equation (2) – (5). We then compared this standard approach to alternative approach 1 where $N_E$ is used but all parameters ($N_0, a_0, a_1, a_2$) from equation (2) – (5) are adjusted. Lastly, we compare the standard calibration approach as well as alternative approach 1 with alternative approach 2. Here, we re-scale the epithermal neutron intensity ($N_E$) by calculating the normalized sum of thermal and epithermal neutrons ($N_{ET}$) based on equation (6) and use the re-scaled neutron intensities in equation (2) – (5) by iteratively calibrating $N_0$. The approaches are applied using a depth-distance weighted reference soil moisture time series as well as the arithmetic average of reference measurements and result in the statistical goodness-of-fit presented in Table 2.





**Table 2.** Statistical goodness-of-fit when calibrating equations (2) – (5) with [applying] the three different calibration approaches.

| Calibration | Reference soil moisture | Neutron intensities | $a_0$ | $a_1$ | $a_2$ | $N_0$ | KGE | NSE | RMSE |
|---|---|---|---|---|---|---|---|---|---|
| Standard | | $N_E$ | 0.0808 | 0.372 | 0.115 | 907.3 | 0.57 | -0.01 | 0.032 |
| Approach 1 | Weighted | $N_E$ | 0.2080 | 0.155 | 0.117 | 595.7 | 0.84 | 0.72 | 0.017 |
| Approach 2 | | $N_{ET}$ | 0.0808 | 0.372 | 0.115 | 956.4 | 0.85 | 0.71 | 0.017 |
| Standard | | $N_E$ | 0.0808 | 0.372 | 0.115 | 896.9 | 0.46 | -0.84 | 0.04 |
| Approach 1 | Arithmetic | $N_E$ | 0.1400 | 0.0083 | 0.103 | 926.4 | 0.84 | 0.61 | 0.019 |
| Approach 2 | | $N_{ET}$ | 0.0808 | 0.372 | 0.115 | 955.4 | 0.79 | 0.61 | 0.019 |

The calibration of the CRNS-derived soil moisture time series by iteratively adjusting $N_0$ based on the standard calibration approach and a weighted reference soil moisture time series results in a KGE of 0.57 (Table 2). When all variables are adjusted
in alternative approach 1 the KGE can be increased to 0.84. Similarly, the KGE increases to 0.85 when alternative approach 2 with re-scaled neutron intensities $N_{ET}$ is applied. This improvement is also expressed by a higher Nash-Sutcliffe efficiency (NSE) and lower root mean square error (RMSE) for the alternative approaches 1 and 2 by either using a re-scaled neutron time series $N_{ET}$ and only adjusting $N_0$, or by using $N_E$ and tuning all parameters. In contrast, using the arithmetic average reference soil moisture time series leads to a lower KGE of 0.46 compared to the calibration against a depth-distance weighted average
with the standard calibration approach. However, the KGE again increases strongly if either alternative approach 1 or alternative approach 2 are optimized against the arithmetic average of reference soil moisture observations to generate a CRNS-derived soil moisture time series. The derived KGE increases to 0.84 and 0.79, respectively, which is close to the goodness-of-fit derived using the weighted reference soil moisture time series. Similarly, the additional goodness-of-fit parameters NSE and RMSE improve when using alternative approaches 1 and 2 instead of the standard approach. In general, for alternative approach 1 and
2, the calibration result in terms of the KGE improves by at least 0.25 over the 4-year study period compared to the standard calibration approach (see also Fig. A1 and Table A1).

The CRNS-derived soil moisture time series based on the three calibration approaches both over- and underestimate the dynamics of the weighted reference time series (Fig. 10). The largest differences from using the standard calibration approach to either tuning all parameters of the transfer function in alternative approach 1 or applying the rescaled $N_{ET}$ in alternative
approach 2 occur in summer periods where the weighted near-field reference soil moisture is generally low. In these periods, using $N_E$ in the standard calibration approach results in a distinct underestimation of the reference soil moisture while both alternative calibration approaches produce a CRNS-derived soil moisture time series fitting the weighted reference more closely (Fig. 11). In contrast, differences between the CRNS-derived soil moisture time series and the reference time series are less pronounced in winter periods when the near-field reference soil moisture is high.
Although differences between the CRNS-derived soil moisture time series vary seasonally, an overall closer fit to the weighted reference time series can be achieved when estimating soil moisture based on the two tested approaches for improving calibration against near-field reference measurements as it was previously illustrated by the statistical goodness-of-fit



**Figure 10.** On-site observed hourly rainfall sums (a) and CRNS-derived soil moisture time series based on the standard calibration approach as well as calibration approach 1 and 2 in comparison to the depth-distance weighted reference soil moisture time series derived from SMT100 sensors (b).

parameters. Lastly, we compared the CRNS-derived soil moisture time series based on the three calibration approaches. A bootstrapping of residuals and subsequent Wilcoxon rank sum tests per time step revealed significant differences ($p < 0.05$)

between CRNS-derived soil moisture time series calculated from the standard calibration approach and the two alternative approaches tested for improving the calibration. For both alternative calibration approaches and both reference soil moisture time series, the CRNS-derived soil moisture time series are significantly different from the time series based on the standard calibration approach for at least 97 percent of the time steps.

**Figure 11.** On-site observed hourly rainfall sums (a) and CRNS-derived soil moisture time series based on the standard calibration approach as well as alternative approach 1 and 2 in comparison to the depth-distance weighted reference soil moisture time series derived from SMT100 sensors (b) for a three month period in the summer of 2016.

## 4 Discussion

### 4.1 Neutron energy-dependent variations of footprint size and neutron intensity

Our neutron transport simulations resulted in footprint radii of epithermal neutrons which lie in the ranges reported by Köhli et al. (2015) and shown in Schrön et al. (2017) for all simulation scenarios. In contrast, little is known about the measurement footprint radius of thermal neutrons. Previous studies assume a similarly sized footprint (e.g., Vather et al., 2020) or a significantly smaller measurement radius (e.g., Bogena et al., 2020). Our simulation results reveal that the derived measurement radii strongly depend on the definition of the point of origin of the thermal neutrons detected by the virtual neutron detector. If the point of thermalization is used to calculate the measurement radius $R_{86}$, the footprint radius has an average size of 49 m and



only exhibits little change to changes in soil moisture. This estimate comes close to what is stated in Bogena et al. (2020). In our simulations with constant soil moisture in near field mineral soils, the fraction of detected neutrons being thermalized in areas covered with mineral soils and peatland soils remains constant although soil moisture in peatland soils is altered (Fig. 5).

This could lead to the interpretation that the footprint of thermal neutrons is too small to cover a significant portion of peatland soils and thus, peatland soil moisture variations do not influence the simulated thermal neutron intensity. However, even if the footprint based on the point of thermalization is small, an influence of peatland soil moisture variations on the amount of thermal neutrons reaching the virtual detector is visible (Fig. 4). As a consequence, the alternative definition may be more suitable: If point of first soil contact is defined as the origin for both, epithermal and thermal neutrons, the thermal neutron

footprint becomes 2-3 three times as large compared to using point of thermalization as the origin. It then covers larger parts of peatland areas and and the fraction of detected thermal neutrons originating from peatland areas changes with varying peatland soil moisture. This definition better explains the variations visible in the detected thermal neutron intensity because the amount of detected thermal neutrons generated from higher-energy neutrons with peatland soil contact is likely to vary with peatland soil moisture.

The average measurement depths of epithermal neutrons simulated here lie also in the range of the values reported in previous studies (Zreda et al., 2008; Köhli et al., 2015). For the case of constant soil moisture conditions in the highly sensitive near-field, the average measurement depth of epithermal neutrons only shows a slight decrease with increasing peatland soil moisture. In the scenarios with varying soil moisture contents in both the near-field and the far-field a change of the CRNS measurement depth is clearly visible. The integration depth of thermal and epithermal neutrons shows a very similar response

to soil moisture variations with thermal neutrons having a slightly shallower $D_{86}$ when based on the point of thermalization. In contrast, if the the maximum depth along the neutron transport path is considered as the measurement depth, the thermal neutron integration depth becomes nearly twice as deep compared to epithermal neutrons. This observation might be explained in the following way: After a high-energy neutron enters the soil column it is slowed down to an epithermal neutron. This epithermal neutron either leaves the soil or is further slowed down to a thermal neutron before then leaving the soil. The

deeper in the soil an epithermal neutron is generated by a high-energy neutron, the more likely it will be thermalized before leaving the soil column. Consequently, thermal neutrons might contain information of soil moisture from even greater depths than epithermal neutrons. However, little is known about the vertical and horizontal footprint dynamics of thermal neutrons in general and care should be taken when interpreting the presented results based on a heterogeneous model domain and a narrow range of simulated boundary conditions. Further research is required to investigate the footprint dynamics of thermal neutrons

for a homogeneous study site and under a wide range of boundary conditions to eventually derive weighting functions similar to those developed for epithermal neutrons during the past decade (e.g., Zreda et al., 2008; Franz et al., 2012; Köhli et al., 2015; Schrön et al., 2017; Scheiffele et al., 2020).

Besides varying dimensions of the integration volume, epithermal and thermal neutrons show a different response to the simulated changes in peatland soil moisture. This applies both to the simulation set with a constant near-field soil moisture and

to the simulation set with the soil moisture in the near-field and in the peatland soils changing at different degrees. In general, the total amount of detected epithermal and thermal neutrons decreases with an increasing peatland soil moisture and thus,





with an increasing amount of hydrogen in the model domain. However, the response of epithermal and thermal neutrons to the different simulated soil moisture contents in mineral and peatland soils differs. Differences in the response of epithermal and thermal neutrons have also been observed in previous modelling studies (e.g., Andreasen et al., 2017b) for variations in soil moisture or liquid water layer thickness (e.g., Hubert et al., 2016). Thermal neutrons show a much smaller response to variations in hydrogen (Weimar et al., 2020) and they show a moderation optimum occurring just below $0.10\,\mathrm{m^3\ m^{-3}}$ soil moisture (Sato and Niita, 2006; Weimar et al., 2020). The latter is caused by the two competing processes influencing thermal neutron abundance: slowing down of epithermal neutrons (moderation) and absorption of thermal neutrons for example by hydrogen (e.g., Hubert et al., 2016).

Several assumption need to be considered when interpreting the neutron transport simulation results presented here. In this study, a simplified model domain was created where topography was neglected. However, while topography may play a more important role in mountainous terrain (e.g., Schattan et al., 2019; Mares et al., 2020), smaller topographic gradients, as at our study site, are unlikely to have a considerable influence on footprint sizes (Köhli et al., 2015). Nevertheless, the slightly elevated position of the observation site compared to the surrounding peatland (Fig. 1) may cause slightly larger real-world footprint radii than those obtained from the simulations. Additional simplifications include the estimation of soil porosities based on the density of quartz and the use of the same soil chemistry for organic and mineral soils. The higher amount of hydrogen stored in the soil organic matter of the peatlands is likely e.g. to shift the simulation results towards lower ratios of epithermal and thermal neutrons and smaller footprint sizes in the real world. As a consequence of the simplifications and limitations of the neutron transport simulations, care needs to be taken when using the model results to explain the real-world observations.

## 4.2 Towards a quantification of footprint heterogeneity

The high correlation between thermal and epithermal neutron intensities at the more uniform sites and the low correlation at the heterogeneous site can be explained by the fact that both energy ranges have different footprint sizes as indicated by Bogena et al. (2020) and partly supported by the neutron simulations shown here. The heterogeneous observation site has a lower correlation coefficient which indicates larger differences between the soil water contents and dynamics of the near field and of far field with organic peatland soils. However, our simulations indicate that the thermal neutron intensity is influenced by far field soil moisture changes occurring in areas covered with peatland soils. The scatter plot for our heterogeneous site A in Fig. 8a shows deviations from the narrow point cloud and thus, close non-linear relationship observed at study sites where soil moisture is more uniform in terms of absolute values and relative changes. The strongest deviations occur at high normalized intensities leading to the heterocedasticity observed and in turn, the lower correlation coefficient. Adding the results from all simulations conducted in the scope of this study to the scatter plot with observed values (Fig. 9) further illustrates this effect. The simulated intensities comprising different absolute soil moisture values for mineral soils of the near-field and peatland soils of the far field as well as different soil water dynamics are located in the area representing deviations from the relationship observed at the two homogeneous observation sites. Especially the simulated normalized numbers of detected neutrons with low static near field soil moisture and changing peatland soil moisture are located in the area of the scatter plot considered as deviations. Under real-world conditions, this response may be expected during summer periods when mineral soil moisture





contents reached a minimum while peatland soil moisture continues to change due to e.g. high water storage capacities and and variations of shallow groundwater visible in Fig. 2 and Fig. 10.

Hence, a twofold influence may be considered. On the one hand, the total soil moisture content differs between the mineral soils of the near-field and the far-field peatland soils leading to a different response of thermal and epithermal neutrons when soil water contents change in the near field and the far field. On the other hand, the differences in moisture content between the near-field and the far-field may fluctuate over time and result in a varying influence on the observed neutron intensities in the two energy ranges.

Against this background, the simple Spearman's rank correlation coefficient can serve as a first indicator for footprint heterogeneity in terms of soil moisture conditions in the near and the far field of the neutron detector. Detailed knowledge of the functional relationship between thermal and epithermal neutrons at heterogeneous observation sites poses great potential for an improved assessment of footprint heterogeneity and the development of advanced indices. However, this requires further research regarding the relationship of thermal and epithermal neutron intensities under changing soil moisture and with respect to different environmental factors such as soil chemistry. This may also require the need to develop transfer functions for thermal neutrons similar to those already available for estimating soil moisture from epithermal neutron intensities (e.g., Desilets et al., 2010; Franz et al., 2013; Köhli et al., 2021).

### 4.3 Improving the estimation of near-field soil moisture

The calibration of neutron observations against near-field reference soil moisture time series from in-situ soil moisture sensors revealed an improvement of the calibration result in terms of the statistical goodness-of-fit when either all parameters of the transfer function are adjusted ($N_0$, $a_0$, $a_1$ and $a_2$) or the combination of thermal and epithermal neutrons based on (6) was used and only $N_0$ was calibrated. Using the standard calibration approach, the calibration against a weighted average reference soil moisture time series resulted in a better KGE compared to a simple arithmetic average. This illustrates the positive effect of the weighting procedure developed by Köhli et al. (2015) and advanced by Schrön et al. (2017) to match the sensitivity of the CRNS.

Both, adjusting all parameters in alternative approach 1 instead of only calibrating $N_0$, as well as combining thermal and epithermal neutrons in the scope of alternative approach 2 leads to an improvement of the calibrated KGE by at least 0.25. As the analyses revealed a significant ($p < 0.05$) difference between the CRNS-derived soil moisture based on the standard calibration approach and both alternative calibration approaches for at least 97 percent of the data points of the time series, a significantly improved representation of the near-field soil moisture dynamics can be achieved by either adjusting all parameters (approach 1) or combining both neutron energy ranges (approach 2). However, this does not represent an uncertainty analysis that considers various statistical sources of uncertainty as it was done in previous studies (e.g., Gugerli et al., 2019; Jakobi et al., 2020).

In this study, reference soil moisture sensors were installed in the near-field of the CRNS only and no information on far-field peatland soil moisture dynamics were available. Although this poses the largest limitation of this study, marked differences of soil water dynamics can be expected for mineral and peatland soils due to the distinct hydraulic behaviour of the latter (e.g.,





Rezanezhad et al., 2016). In addition, peatland areas at the study site are characterised by groundwater influence and potentially higher soil water content, as observed in the reference soil samples taken from mineral and peatland soils in February 2020 (section 2.1). The obtained calibration results of CRNS-derived soil moisture time series clearly show an improvement of the representation of near-field soil moisture dynamics when accounting for these peatland soil moisture dynamics through alternative approach 1 or 2.

The transfer function developed by Desilets et al. (2010) and revised by Köhli et al. (2021) was designed for a uniform soil water content within the measurement footprint. In the example presented in this study, the epithermal neutron response is stronger than the soil water changes observed in mineral soils of the near-field causing the underestimation of near field reference soil moisture during summer periods. Therefore, adjusting the shape-defining variables of eq. (2) – (5) in addition to the calibration parameter $N_0$ alone in alternative approach 1 allows for adjusting the transfer function for the soil moisture

dynamics of the near field. Different studies already adjusted the shape-defining parameters of the standard calibration function (eq. (1)) (e.g., Heidbüchel et al., 2016) and achieved better calibration results compared to adjusting $N_0$ only. However, reasons for changing the physical meaning of the eq. (1) or (eq. (2) – (5)) by tuning all variables of the transfer function remain disputable. The results of this study shed more light on potential reasons for an improved calibration against reference measurements by changing the shape of the transfer function: Distinct differences of soil moisture states and dynamics within

the measurement footprint over time may lead to neutron responses deviating from the shape of the original transfer function (eq. (1) or (eq. (2)). This is in line with findings from previous studies (e.g., Lv et al., 2014; Heidbüchel et al., 2016). Nevertheless, care should be taken when tuning the shape-giving parameters for optimizing the goodness-of-fit against different reference measurements. This can lead to different optimized values for the shape-giving parameters ($N_0$, $a_0$, $a_1$ and $a_2$) depending on the reference soil moisture time series (see also Table A1). This illustrates that the shape-giving parameters can be

fitted to different reference measurements and that the objective for the site-specific optimization needs to be considered. For example, optimizing for a site-specific areal average requires sufficient spatio-temporal coverage of reference measurements while optimizing for reference soil moisture in defined parts of the footprint requires representative reference measurements from these areas.

In contrast, in alternative approach 2, we produced a rescaled neutron time series $N_{ET}$ based on observed epithermal ($N_E$)

and thermal $N_T$ neutron intensities and thus, adjusted the signal instead of the transfer function. As the thermal neutron response to soil water changes is generally weaker (Weimar et al., 2020), summing the observed normalized intensities of thermal and epithermal neutrons and rescaling them using eq. (6) leads to a less steep slope of the functional relationship between neutron intensity and reference soil moisture. Consequently, the rescaling approach presented here makes use of thermal neutrons ($N_T$) as a proxy for a different response to soil water changes leading to a rescaled neutron time series $N_{ET}$ that is more similar

to epithermal neutron intensities $N_E$ if the entire measurement footprint having the lower soil water content and dampened dynamics of the near-field where the reference point sensors are installed. Hence, the second alternative approach tested in this study is suitable for a separation of near and far field soil moisture at the heterogeneous observation site investigated.

Nevertheless, limitations need to be considered when assessing the improvement achieved with the different approaches tested in this study. Besides the horizontal footprint of thermal neutrons being smaller than for epithermal neutrons, the in-





tegration depth can be considered to be different as well. This may complicate the joint interpretation and combination of both neutron energies and intensities. Additionally, the differing influences of other factors on thermal and epithermal neutron observations should be considered. In this study, raw thermal and epithermal neutron observations were corrected equally for the influence of variations in atmospheric shielding depth and incoming high-energy neutron radiation, but not for variations in air humidity as the latter was determined for epithermal neutrons only. The correction procedures applied to observed thermal

neutrons differ among studies (e.g., Andreasen et al., 2016; Jakobi et al., 2018) and in addition to the need of detailed knowledge about the dynamics of the integration volume, further research is required concerning appropriate correction procedures for thermal neutrons to varying environmental conditions.

## 5 Conclusions

The neutron transport simulations performed here support previous studies in indicating a distinctively smaller horizontal
measurement footprint of thermal compared to epithermal neutrons. However, the thermal neutron footprint radius strongly depends upon the definition of the origin of detected neutrons. Our study suggests that the point of thermalization alone may not be suitable for characterising the sensitive measurement footprint size as detected thermal neutrons do vary with far field soil moisture variations. Instead, equally to epithermal neutrons, the point of first soil contact may be more suitable. In this case the the integration radius almost doubles, but still remains smaller than that of epithermal neutrons. The integration depth
also increases strongly, even surpassing that of epithermal neutrons.

The relationship between normalized observed thermal and epithermal neutron intensities is likely to differ between homogeneous and heterogeneous conditions and may be used to characterise footprint heterogeneity. The simple Spearman's rank correlation coefficient between the normalized thermal and epithermal neutron intensities proved to a be a suitable first indicator for the footprint heterogeneity with lower values indicating a larger (as well as varying) difference between soil water
contents in the near- and the far field.

Either adjusting all parameters of the transfer function or rescaling observed epithermal neutron intensities by averaging the normalized dynamics of thermal and epithermal neutrons leads to a significant improvement of the calibration result against reference soil moisture sensors in the near-field. This is achieved by changing the neutron intensity dynamics towards a more dampened response that would occur if the entire epithermal footprint showed the lower soil moisture conditions and dampened
dynamics of the near-field.

In conclusion, both approaches tested for improving the estimation of near-field soil moisture pose great value for the use of CRNS at study sites with heterogeneous soil water contents and dynamics. On the one hand, complementary observations of thermal and epithermal neutrons offer the opportunity to test for footprint heterogeneity using simple correlation measures. On the other hand, in addition to adjusting the transfer function, the thermal neutron intensity proved to be a useful proxy for
rescaling the epithermal neutron intensities in order to improve the representation of near-field soil water time series at the study site. Several limitations of this study need to be considered and illustrate the need for further research, especially regarding the general response of thermal neutrons to environmental conditions, suitable correction procedures for these phenomena, as well





as the behaviour of both neutron energies at study sites with heterogeneous distributions of soil water and pools of hydrogen in general. Nevertheless, this study illustrates possibilities of a spatial discretization of soil moisture at heterogeneous study sites

and the potential of using both neutron energies for improving CRNS-derived soil water estimates.

*Data availability.*  Data sets of the all CRNS sensors will be published through GFZ Data Services (https://dataservices.gfz-potsdam.de/portal/) and will also be made available through the TERENO data portal (www.tereno.net). Until then, data sets are available from the authors upon request.

**Figure A1.** On-site observed hourly rainfall sums (a) and CRNS-derived soil moisture time series based on the standard calibration approach as well as calibration approach 1 and 2 in comparison to the depth-distance weighted reference soil moisture time series derived from TDR sensors (b).





**Table A1.** Statistical goodness-of-fit when calibrating equations (2) – (5) with [applying] the three different calibration approaches using reference soil moisture observations from TDR sensors with a higher noise and lower signal quality.

| Calibration | Reference soil moisture | Neutron intensities | $a_0$ | $a_1$ | $a_2$ | $N_0$ | KGE | NSE | RMSE |
|---|---|---|---|---|---|---|---|---|---|
| Standard | | $N_E$ | 0.0808 | 0.372 | 0.115 | 849.5 | 0.40 | -0.58 | 0.040 |
| Approach 1 | Weighted | $N_E$ | 0.1420 | 0.153 | 0.100 | 724.6 | 0.80 | 0.64 | 0.019 |
| Approach 2 | | $N_{ET}$ | 0.0808 | 0.372 | 0.115 | 899.9 | 0.68 | 0.31 | 0.026 |
| Standard | | $N_E$ | 0.0808 | 0.372 | 0.115 | 842.6 | 0.33 | -1.03 | 0.044 |
| Approach 1 | Arithmetic | $N_E$ | 0.1420 | 0.153 | 0.100 | 724.6 | 0.78 | 0.58 | 0.020 |
| Approach 2 | | $N_{ET}$ | 0.0808 | 0.372 | 0.115 | 898.4 | 0.64 | 0.19 | 0.028 |

*Author contributions.*  DR designed the study, performed the data analysis and wrote the manuscript. AG and TB designed the experimental

network and contributed to the writing of the manuscript. MK and MS assisted in performing and analysing neutron transport simulations and reviewed the manuscript.

*Competing interests.*  The authors declare no competing interests, however, TB is Chief Executive Editor of the journal.

*Acknowledgements.*  This study was conducted as part of the research unit CosmicSense funded by the German Research Foundation

(Deutsche Forschungsgemeinschaft, DFG-FOR2694). We gratefully acknowledge the technical support of Markus Morgner, Jörg Wummel and Stephan Schröder who maintain the observation sites in TERENO-NE funded by the Helmholtz Association. In addition, we would like to thank the Stadtwerke Neustrelitz GmbH for supplying on-site time series of groundwater levels as well as Peter Stüve and Paul Voit for their assistance in data acquisition, field and laboratory work. Further, we would like to thank the Müritz National Park and local landowners for their continuing support and collaboration. Lastly, we acknowledge the NMDB database (www.nmdb.eu) founded under the European

Union's FP7 programme (contract no. 213007), and the PIs of individual neutron monitors for providing data.



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
