# Peer review of "Towards disentangling heterogeneous soil moisture patterns in Cosmic-Ray Neutron Sensor footprints"

_Hydrology and Earth System Sciences, 2021_

## Referee Comment (RC2)

Review of Towards disentangling heterogeneous soil moisture patterns in Cosmic-Ray Neutron Sensor footprints by Rasche et al.

**Summary**

The Authors explored the use of thermal and epithermal neutrons to detect soil moisture spatial variability within a CRNS sensor footprint. The study is based on several simulations performed with URANOS model and long term experimental data collected at one location in Germany. The manuscript is well structured and written, results are clearly explained and discussed. In my opinion the manuscript is suitable for the journal but should be strengthen in several parts. Below I provide my general concerns followed by specific comments in order of appearance in the manuscript. I hope these comments could help for further improvements.

**General comments**

[1]. The more complex response of the thermal neutrons rise my main concern of the present study. The potential use of thermal and epithermal neutrons has been pointed since the first CRNS publications (Desilets et al., 2010; Rivera Villarreyes et al., 2011). Further attempts have also been performed later by dedicated studies (Bogena et al., 2020; Jakobi et al., 2018; Tian et al., 2016). As far as I understood, difficulties to handle these two signals are related to the non-unique response of the thermal neutrons, i.e., in contrast to epithermal, they depend on chemistry and the thermal intensity also increases during the wetting of initially dry soils (Desilets et al., 2010; Zweck et al., 2013). The present study quantifies the different footprint of thermal and epithermal neutrons. As such, it sheds lights on the understanding on the processes. However, few is discussed on the possibility to generalize the correlation found in the present study between epithermal and thermal in other conditions. As such I see the high risk of this study to be very limited. In addition, if the Authors are really interested on disentangling footprint variability, I rather believe that the use of side-shielded detector (Zreda et al., 2021) could be easier and more promising than the use of the thermal detector.

[2]. As far as I have understood, the comparison between neutrons simulations and neutron measurements is not consistent. Simulations are based on theoretical detectors sensitive only to thermal or epithermal ranges. In contrast, measurements have been collected with bare and moderated detectors that are contaminated by epithermal and thermal neutrons, respectively, as highlighted by the Authors. Previous studies showed clear discrepancy between simulations and measurements when this contamination effect was not properly account for (Andreasen et al., 2016; McJannet et al., 2014). As such, I'm surprised about this setting. Either the detectors should be improved to remove the contamination from thermal and epithermal. If this is not possible within the present study, why not repeating the simulations with the real detectors? Despite more rigorous understanding of thermal and epithermal, you are allowed to compare the simulations and the experimental data.

[3]. Point scale soil moisture observations are very limited and they represent only short distance. This has been pointed as main limitation of the present study but very late in the manuscript and without explaining the consequences of that. Please note that the use of limited number of soil moisture locations have been highly criticized in former studies (see discussion for (Rivera Villarreyes et al., 2013). Despite I'm personally do not against comparisons with relative few points sensors, it should be noted that the present study concluded that the use of thermal signal improved the performance. However, I see a strong bias if we consider that thermal has a smaller footprint and point scale soil moisture used for the comparisons are located in the near field. As such, I would rather presume to have worst results in case the point soil moisture sensors would have been distributed also at larger distance.

**Specific comments**

L2: I suggest the term estimations instead of measurements, i.e., the sensors measures neutrons and, based on that, estimate soil moisture.

L4-5: I think it should better phrased. 1) soil homogeneous conditions are unlikely and from my understanding 2) the added value of CRNS emerges exactly in case of heterogeneous conditions. The key assumption is in my opinion to sense a representative volume where soil moisture shows a relative short correlation length. In this case neutrons well mix within the footprint. In case of longer correlation length and spatial patterns, empirical data deviates from theoretical functions and hysteresis behavior could also emerge. Similar consideration has been detected in snow patches conditions (Schattan et al., 2019)

L39. If I'm not wrong, some papers refer to the threshold 0.5 for thermal neutrons. Could you provide reasoning for this value?

L43. The more complex response of the thermal neutrons rise my main concern of the present study, i.e., the results are very site specific (see general comment above)

L65. I'm very surprised if most studies with stationary CRNS assume homogeneous site conditions. Please rephrased as previously discussed.

L105. Please add if possible the 21 random locations on figure 1.

L125. but why to simulate something that it does not represent the real detector? See general comment above

L131. point scale soil moisture observations are very limited, they represent only short distance and they are not evenly distributed. The CRNS calibration is strongly biased

L166-183. This text refers to all simulations and not only to simulation set 1. It should be moved up in section 2.2.

L180. As far I understood from previous studies, D86 is not spatially constant. Please specify if you refers here to the maximum (or average) depth over the footprint

Figure 2. ground water level should be reported as depth from soil surface to ground water instead of ground water level above sea level to facilitate the interpretation on the discussion on shallow water table influencing soil moisture detected by CRNS.

Equation 6: you merge thermal and epithermal with different footprints. But you compare the scaled sum with point scale soil moisture weighted based on epithermal footprint. Are you not mixing up the signals? Additionally, what about using only thermal? I expect good or even better results when calibrating with these near field point locations.

Figure 3b. If you calculate the maximum D86, it should be expected to not changing much the depth by increasing far-field soil moisture. Please clarify

**References**

Andreasen, M., Jensen, K.H., Zreda, M., Desilets, D., Bogena, H., Looms, M.C., 2016. Modeling cosmic ray neutron field measurements. Water Resour. Res. 52, 6451–6471. https://doi.org/10.1002/2015WR018236

Bogena, H.R., Herrmann, F., Jakobi, J., Brogi, C., Ilias, A., Huisman, J.A., Panagopoulos, A., Pisinaras, V., 2020. Monitoring of Snowpack Dynamics With Cosmic-Ray Neutron Probes: A Comparison of Four Conversion Methods. Front. Water 2. https://doi.org/10.3389/frwa.2020.00019

Desilets, D., Zreda, M., Ferré, T.P.A., 2010. Nature's neutron probe: Land surface hydrology at an elusive scale with cosmic rays. Water Resources Research 46. https://doi.org/10.1029/2009WR008726

Jakobi, J., Huisman, J.A., Vereecken, H., Diekkrüger, B., Bogena, H.R., 2018. Cosmic Ray Neutron Sensing for Simultaneous Soil Water Content and Biomass Quantification in Drought Conditions. Water Resour. Res. 54, 7383–7402. https://doi.org/10.1029/2018WR022692

McJannet, D., Franz, T., Hawdon, A., Boadle, D., Baker, B., Almeida, A., Silberstein, R., Lambert, T., Desilets, D., 2014. Field testing of the universal calibration function for determination of soil moisture with cosmic-ray neutrons. Water Resources Research 50, 5235–5248. https://doi.org/10.1002/2014WR015513

Rivera Villarreyes, C.A., Baroni, G., Oswald, S.E., 2013. Calibration approaches of cosmic-ray neutron sensing for soil moisture measurement in cropped fields. Hydrology and Earth System Sciences Discussions 10, 4237–4274. https://doi.org/10.5194/hessd-10-4237-2013

Rivera Villarreyes, C.A., Baroni, G., Oswald, S.E., 2011. Integral quantification of seasonal soil moisture changes in farmland by cosmic-ray neutrons. Hydrology and Earth System Sciences 15, 3843–3859. https://doi.org/10.5194/hess-15-3843-2011

Schattan, P., Köhli, M., Schrön, M., Baroni, G., Oswald, S.E., 2019. Sensing Area-Average Snow Water Equivalent with Cosmic-Ray Neutrons: The Influence of Fractional Snow Cover. Water Resour. Res. 55, 10796–10812. https://doi.org/10.1029/2019WR025647

Tian, Z., Li, Z., Liu, G., Li, B., Ren, T., 2016. Soil Water Content Determination with Cosmic-ray Neutron Sensor: Correcting Aboveground Hydrogen Effects with Thermal/Fast Neutron Ratio. Journal of Hydrology. https://doi.org/10.1016/j.jhydrol.2016.07.004

Zreda, M., Hamann, S., Schrön, M., Köhli, M., 2021. Distance and direction-sensitive cosmogenic neutron sensors. US20210102906A1.

Zweck, C., Zreda, M., Desilets, D., 2013. Snow shielding factors for cosmogenic nuclide dating inferred from Monte Carlo neutron transport simulations. Earth and Planetary Science Letters 379, 64–71. https://doi.org/10.1016/j.epsl.2013.07.023

---

## Author Comment (AC1)

**Reviewer #1:**

We thank reviewer #1 for taking the time to review our manuscript and their valuable suggestions regarding our manuscript. We are certain that these comments greatly improve our manuscript and they will be incorporated in a revised version of the manuscript. In the following section we will reply to all comments of reviewer #1 with R1-1 (i.e. reviewer 1, comment 1) and A1-1 (i.e. author response to R1-1), respectively.

**R1-1:** The figure legends and axis labels are too small. Please enlarge before publication.

**A1-1:** We will enlarge axis labels and legend labels of figures in a revised version of our manuscript.

**R1-2:** L444. The authors argue that the thermal neutron footprint may be significantly deeper than the epithermal range of ~30-40 cm. If this is indeed the case additional profile sampling of soil chemistry is needed by the community in order to understand the distribution of trace elements (e.g. Gd and B) that may greatly impact the thermal neutrons. In particular, as the neutrons interact with more soil horizons (beyond the O and A typical for epithermal) soil chemistry may play a greater role. The authors point this out a little but should highlight the need by the community to sample more soil horizons for relevant epithermal and thermal neutron soil chemistry. Avery et al. 2016 and others have presented a nice lattice water dataset for the top 30 cm but it seems the community needs to expand this effort across CRNS sites and more soil horizons.

**A1-2:** The integration depth of thermal neutrons remains under debate. We found that the average measurement depth ($D_{86}$) strongly depends upon the definition of integration depth. If the point of thermalization (i.e. the point where a neutron first reaches a thermal energy) is used, the measurement depth is slightly shallower then the one of epithermal neutrons while it becomes much larger when the maximum depth along the complete neutron transport path is used. We intended to show that more research is needed in order to understand the thermal neutron transport under different environmental settings and cannot deliver a final solution on the definition of the integration depth of thermal neutrons. However, we will add a sentence regarding the potential implications of a larger thermal measurement depth with respect to soil sampling and information on soil chemistry to a revised version of this manuscript.

**R1-3:** L475. The authors show the heteroscedasticity effect from local and far-field soil moisture changes on the thermal and epithermal scatterplots nicely in Fig 8A. Without the soil moisture data in the peatlands, the conclusion is somewhat more speculative based on GW depth but still compelling. However, additional CRNS sites with largescale irrigation (60 ha) from

center pivots may confirm this effect (CRNS sites exist in NE, KS, and IA in the USA with center pivots). As the center pivots water in pie slices over 48-72 hours they will create this near and far-field effect, particularly when compared against a rainfall event on-site. The authors could mention this experiment as future work needed by the CRNS community to help confirm the conclusions here.

5

**A1-3:** We agree with the reviewer, and had in fact already included a similar idea in a proposal. Accompanied with intensive soil sampling or in-situ sensor networks as well as neutron transport simulations, such experiments could also help in finding supporting evidence for the measurement radius of thermal and epithermal derived from neutron transport simulations in numerous studies during the past decade. For instance, and with respect to this study, such an experimental setup could help

10    finding an answer of whether using the point of thermalization, the point of the first soil contact or a different characteristic of a simulated detected thermal neutron for calculating the footprint radius. We will add this information at the end of chapter 4.2 (L. 496) of a revised version of this manuscript as stated below.

Original: "However, this requires further research regarding the relationship of thermal and epithermal neutron intensities

15    under changing soil moisture and with respect to different environmental factors such as soil chemistry. This may also require the need to develop transfer functions for thermal neutrons similar to those already available for estimating soil moisture from epithermal neutron intensities (e.g., Desilets et al., 2010; Franz et al., 2013; Köhli et al., 2021)."

Adjustment: "However, this requires further research regarding the relationship of thermal and epithermal neutron intensities

20    under changing soil moisture and with respect to different environmental factors such as soil chemistry. As study sites are always restricted to local boundary conditions, large scale irrigation experiments using e.g. center pivots combined with neutron transport simulations could improve and extend the insights gained in this study regarding indicators for footprint heterogeneity as well as the definition of measurement footprints in general. Additionally, this may also require the need to develop transfer functions for thermal neutrons similar to those already available for estimating soil moisture from epithermal

25    neutron intensities (e.g., Desilets et al., 2010; Franz et al., 2013; Köhli et al., 2021)."

**R1-4:** Need space "time seriesNET"

**A1-4:** We will add a space.

30

**R1-5:** Figure 3. check legend and data time series in panel B? No red squares etc.

**A1-5:.** In figure 3, panel b), the colours are correct but indeed the point symbols are not. We apologize for this mistake and will correct it.

**R1-6:** L538. Also, watch out for overfitting if only calibration data is available on campaign days. The CRNS community has found ~3 calibration dates are needed for robust N0 estimation. For 4 parameters you may need 12 or more calibration sampling days. That is a lot of digging :). It is already challenging to calibrate on multiple days if you have several CRNS sites.

5

**A1-6:** We agree with the reviewer. As stated in the manuscript (L. 540), adjusting the parameters requires a sufficient spatio-temporal coverage of a sufficient number of reference measurements. In this study, permanently installed point sensors were used which provide continuous data and thereby support the calibration process. We believe that further soil sampling campaigns would be time-consuming, labour-intensive, and not add to the already continuous dataset.

10

---

## Author Comment (AC3)

We thank reviewer #3 for taking the time to review our manuscript and their valuable suggestions regarding our manuscript. We are certain that these comments greatly improve our manuscript and they will be incorporated in a revised version of the manuscript. In the following section we will reply to all comments of reviewer #3 with R3-1 (i.e. reviewer 3, comment 1) and

5 A3-1 (i.e. author response to R3-1), respectively.

**R3-1:** Fig 6a: Why is there so little change in the R86 footprints in this figure (as measured by first soil contact) compared to Fig 3a? In the case of simulation set 3 (dashed lines Fig 6a) far field soil moisture drops from 0.7 to 0.2 which is very similar to the drop from 0.7 to 0.1 in Fig 3, but in Fig 6 we also have that the near field soil moisture is decreasing. Surely there should

10 therefore be a larger change in the various footprints in Fig 6? In fact R86 actually decreases slightly in simulation set 2 between scenario 1 (wettest) and scenario 6 (driest)! Is there a mistake? Have missed something important?

**A3-1:** Thank you for this comment! We checked the postprocessing procedure of the simulation data of all simulation scenarios conducted and found a mistake in the code. Fixing the code leads to different absolute simulated neutron intensities and radial

15 footprints. For instance, the average footprint radius of epithermal neutrons in simulation set 1 decreases from 134 to 121 m as we now also use the detector boundary instead of the detector centre for calculating the distances to the point of origin of each neutron. Although the absolute values are different, in relative terms, the results are very similar to those already in the manuscript and thus, fixing the code does not lead to a different interpretation of the simulation results. This also confirms the low sensitivity of our results to the site-specific footprint size. Nevertheless, we apologize for this mistake and will update all

20 values throughout the manuscript and update all figures. The updated figures are shown below.

In respect to Figure 6, correcting the code does not have a major effect. First of all, simulating a relatively small virtual detector in a large model domain leads to a limited number of simulated neutrons actually reaching the neutron detector compared to e.g. measuring the total number of neutrons in a certain energy range are reflected from the soil in the entire model domain

25 which cannot be done under heterogeneous soil moisture conditions. Consequently, a certain degree of statistical variations has to be considered. This is also displayed in Figure 4 and can explain certain variations visible also in the figures showing the calculated $R_{86}$ and $D_{86}$. This is likely to be the most important reason for the slight decrease and variations visible for the $R_{86}$ in fig. 6a for epithermal neutrons.

The footprint change of epithermal neutrons differs between the simulation set 1 and those of simulation set 2 and 3. This can

30 also be related to the fact of the footprint change becomes smaller with more water (i.e. hydrogen) being in the model domain. Due to this non-linearity, the largest footprint changes can be expected under rather dry conditions from e.g. 0-0.15 $m^3$ $m^{-3}$. As a consequence, one reason for the small differences in footprint changes can be related to the high soil moisture contents simulated and adds to the statistical uncertainty of the Monte-Carlo simulations mentioned before which has a higher impact if the footprint change with changing soil moisture is smaller.

When more water is located close to the detector, where it is most sensitive, fewer neutrons reach the detector and the statistical noise increases. However, simulating a discrete virtual neutron detector requires a very high number of source neutrons to be simulated and is computationally intensive. This study is a first investigation of the influence of soil moisture patterns on the radial footprint sizes and thus, a full and general analysis of the measurement footprint of thermal and epithermal neutrons is

5 beyond the scope of this study, but underway for thermal neutrons in a recent preprint of some of the authors. Despite the limitations regarding the accuracy of the simulation results compared to the real-world site, they still allow valuable conclusions within the scope of this study, that e.g. thermal neutrons have a smaller footprint radius, that far-field soil moisture variations still have an influence on the thermal neutron count rate and thus, that the definition of the origin for calculating the footprint radius (e.g. point of thermalization or point of first soil contact) needs to be investigated further – especially under

10 heterogeneous distributions of soil water in the model domain. Lastly, the influence of the geometry and spatial distribution of hydrogen (-variations) in respect to the detector location remains largely unknown and could also be a reason for the partly inconclusive behaviour of calculated radial measurement footprints for the simulations with an equal decrease of soil moisture. The anisotropy of CRNS footprints was already described in Schattan et al. (2019).

15 Updated figures:

[Figure]

**Figure 3**. Simulation results for the measurement footprint radius (a) and depth (b) of detected thermal and epithermal neutrons.

[Figure]

**Figure 4**. Total number of neutrons in the thermal and epithermal energy range observed by the virtual detector per simulated peatland soil moisture.

[Figure]

**Figure 5**. Fraction of detected epithermal and thermal neutrons with increasing soil moisture originating from areas covered with peatland soils and mineral soils in the model domain. For epithermal neutrons, the point of origin is defined as the point of first soil contact while for thermal neutrons both calculations, for the point of first contact and the point of thermalization are shown.

[Figure]

**Figure 6**. Simulated measurement footprint radius R86 (a) and depth D86 (b) of thermal and epithermal neutrons when soil moisture in areas with mineral and peatland soils decreases by the same amount (solid lines), and when peatland soil moisture decreases twice as much (dashed lines).

[Figure]

**Figure 7**. Simulated normalized thermal and epithermal neutron response when soil moisture in areas covered with mineral and peatland soils decreases in equal intervals (solid lines) and when peatland soil moisture decreases twice as much (dashed lines).

[Figure]

**Figure 9**. Relationship between normalized thermal and epithermal neutron intensities for in-situ observations and relationship between the normalized detected epithermal and thermal neutrons for simulated data. The simulated values refer to the simulation set and scenarios summarised in Table 1. Simulated neutrons are normalized by the average number of detected neutrons of all simulations in the respective energy range.

**R3-2:** Footprints: One question that remains in my mind is the practical relevance of simulated $R_{86}$ and $D_{86}$ footprints. This is especially the case for the thermal neutrons where the authors explicitly consider different definitions for the distance travelled by an individual neutron. But even for an epithermal neutron a choice is made to measure distance from the first interaction with the soil, rather than for example some weighted average of the distances from all interactions with the soil. This isn't a criticism particular to this manuscript - it is a general practice when simulating $R_{86}$ for epithermal energies.
One might hope that the $R_{86}$ footprint would approximately have something like the following property,

$$N = 0.86*(p1*N1 + (1-p1)*N2) + 0.14*(p2*N1 + (1-p2)*N2 )$$

where N is the counts detected at the detector, N1 and N2 are the counts that would be detected if the entire area was mineral soil or peat soil respectively (with their own VWC), and p1 and p2 are the proportion of the landscape from within or outside of the R86 distance respectively that is mineral soil. This kind of reasoning is already alluded to around lines 425. But perhaps

this can be quantified maybe using something like the equation above? Perhaps p1 and p2 could be estimated? N1 and N1 could be added to Fig 4? Would similar hold for the both thermal and epithermal footprints? One could even envisage using the above equation as a definition for a footprint radius if a simplified circular geometry (p1=1, p2=0) was employed for the mineral soil. In any case extra discussion would be helpful.

**A3-2:** Thank you for this interesting comment and like the idea of the proposed equation. Parameters p1 and p2 in the proposed equation would change with soil moisture content which requires them to be estimated for each combination of soil water contents in mineral and peatland soils. More importantly, we consider the radial measurement footprint in our study but the real footprint is likely to be rather anisotropic at study sites with distinct different soil moisture patterns. This was already

10 mentioned by Schattan et al. (2019) for partly snow-covered conditions. This does not falsify the equation above but complicates its application and the estimation of the parameters p1 and p2 for different site conditions. Additionally, we did not conduct simulations with the porosity and soil moisture for either mineral or peatland soils in the entire model domain. Performing further simulations with these set ups and several moisture contents is beyond the scope of this study. Nevertheless, we agree, that this requires further investigation and should be addressed in future studies. This is pointed out by the reviewer

15 when mentioning that different definitions of the neutron origin or the $R_{86}$ might be considered. The definition of the origin of a neutron in the model domain remains under debate and different definitions might even be more suitable. In the scope of dedicated footprint studies the above described equation could be tested when the most suitable footprint definition has been found. This also leads to irrigation experiments using centre pivots mentioned by reviewer #1 which could assist in evaluating results from neutron transport simulations.

20

**R3-3:** Fig 3: Perhaps a comment on why R86 for thermal neutrons as measured from the first soil contact isn't in fact larger than R86 for the epithermal neutrons. I could imagine that as an epithermal neutron undergoes further collisions it will eventually reach thermal energies and will had further opportunity to travel from its initial soil contact – although I appreciate

25 the picture is not be as simple as this.

**A3-3:** Thank you for this comment. We think that, when the point of first soil contact is used, the smaller footprint of thermal neutrons compared to epithermal neutrons might also be linked to the deeper integration depth. A secondary neutron generated from a high energy neutron in the soil via nuclear evaporation it is more likely to escape the soil as a neutron with less energy

30 when it was generated in deeper layers due to more scatterings in the soil. When leaving the soil, the travel distance is then limited due to its lower energy. However, we agree that this is only one possible explanation and cannot be fully assessed within the scope of this manuscript. This again illustrates the need for more dedicated future footprint experiments either by simulations or in the field.

**R3-4:** Equation 6: This is an equally weighted normalised average of NT and NE. But its not clear at this point why this is done. It is explained that this combination makes the response have a "shallower slope" than NT, but one normally expects reduced sensitivity to be a bad thing! Perhaps the actual reason is a compromise between having a footprint more representative of the location in which the soil moisture sensors are installed (NT), and the better sensitivity of NE? There is additional explanation around line 545. Also, when using this "alternative approach 2" perhaps one needs to recalibrate the parameters a0, a1, a2, as I believe the original choice of these was made for the epithermal neutrons?

**A3-4:** Reviewer #2 made a similar comment (see also R2-15 and A2-15). The rescaled signal based on equation 6 is more similar to the theoretical epithermal intensity occurring if the entire footprint would have the soil moisture conditions of the near-field of the sensor and thus, better matching the shape of the functional relationship of the standard transfer function developed for epithermal neutrons.

One reason for is the smaller footprint of thermal neutrons more likely to less influenced by far-field soil moisture variations but then requires the better sensitivity of epithermal neutrons as pointed out by reviewer #3.

A more important reason for the improvement is the generally smaller decrease of thermal neutrons with increasing soil moisture. As a consequence, we can make use of thermal neutrons as proxy for a different signal response better matching the response which would occur if the soil moisture conditions of the near-field would cover the entire footprint at our site.

As described in the responses to reviewer #1 and #2, this approach may not be directly transferable to sites with different spatial patterns of soil moisture and different dynamics. However, the more general approach by adjusting the transfer function instead of adjusting the neutron signal could be used instead. This will be more strongly emphasized in the revised version of our manuscript.

**R3-5:** Fig 3b: There's a problem with the legend.

**A3-5:** Thank you, we will correct this in the revised version of the manuscript.

**R3-6:** Lines 16 and 84: "spatial discretization" is supposed to be "spatial disaggregation"? Also line 84 could be clearer.

**A3-6:** We will replace "spatial discretization" with "spatial disaggregation" as suggested.

**R3-7:** Section 2.2.1: Some of the details in this section are general to all simulations (e.g. the detector radius, the energies of the thermal/epithermal neutrons) and would therefore be better in section 2.2.

**A3-7:** This was already mentioned by reviewer #2. We will shift lines 166-183 to section 2.2, insert them after line 149 and make it a bit clearer.

**R3-8:** Fig 5: I can't really see the reason to show both the blue lines and the green lines – they sum to 1.

**A3-8:** Yes, the lines sum to 1. We think it makes it easier to see how many neutrons originate from either region (peatland or mineral soils) when both lines are shown.

**R3-9:** Fig 6: Could be more easily understood if x-axis labelled by near field soil rather than scenario. This is especially because when reading the x-axis left to right it becomes drier which is the opposite way around compared to Fig3.

**A3-9:** We agree with the reviewer on the improvements on this figure. Using the simulated soil moisture in the near-field instead of the scenario number does help interpreting the figures. We will modify figures 6 and 7 accordingly.

**R3-10:** Fig 9: I can understand the authors might be pleased with this figure but why not simply add the simulation points to Fig 8a instead.

**A3-10:** We decided to use a separate figure because we find that merging the information with Figure 8a would make the latter difficult to read and visually separate the data points based on the neutron simulations. We would also prefer to keep Figure 8 purely measurement based instead of mixing in simulations here.

**R3-11:** Line 433: Add reference to the Figure.

**A3-11:** We will add the reference to figure 4.

**R3-12:** Line 272: I think the bandwidth should have time/frequency units? Partly I ask because, I think that if the smoothing is too intense your residual "noise" will actually contain some of the soil moisture signal. I therefore want a rough idea how much smoothing occurred. Not that I think excess noise causes a problem, given the result stated on line 512. And I never doubted that the different approaches where significantly different.

**A3-12:** The smoothing bandwidth of 1,000 in the Nadaraya–Watson kernel smoother does indeed lead to an intense smoothing effect. It keeps the seasonal variations of soil moisture and does also remove some of the soil moisture dynamics in order to generate large residuals for subsequently generating random value distributions per time step which can be compared between the different approaches. We do agree that the selection of the bandwidth is somewhat speculative and an inherent limitation of this time series comparison. We will add this information to the methods section in the manuscript.

**References**

Schattan, P., Köhli, M., Schrön, M., Baroni, G., and Oswald, S. E.: Sensing Area-Average SnowWater Equivalent with Cosmic-Ray Neutrons: The Influence of Fractional Snow Cover, Water Resources Research, 55, 10 796–10 812, https://doi.org/10.1029/2019wr025647, 2019.

---

## Author Comment (AC4)

**Reviewer #4:**

We thank reviewer #4 for taking the time to review our manuscript and their valuable suggestions regarding our manuscript. We are certain that these comments greatly improve our manuscript and they will be incorporated in a revised version of the manuscript. In the following section we will reply to all comments of reviewer #4 with R4-1 (i.e. reviewer 4, comment 1) and

5   A4-1 (i.e. author response to R4-1), respectively.

**R4-1:** Fig 8 is powerful in demonstrating how this ratio between epithermal and thermal neutrons can change at sites with known heterogeneous soil moisture dynamics. I can see this being a useful metric to describe CRNS site heterogeneity in a simple way that can help a user understand possible site-specific impacts on soil moisture dynamics. I wonder if we should

10   reconsider sensor footprint size once the Spearman rank correlation coefficient falls below a certain value?

**A4-1:** This is an interesting comment! We agree that the radial footprint size and anisotropy should be assessed in more detail if distinct differences of soil moisture patterns and dynamics occur within the expected measurement footprint (i.e. 200 m) as this will lead to deviations from existing footprint definitions which are based on neutron transport simulations with a

15   homogeneous soil moisture distribution in the model domain. Defining a threshold value for the Spearman's rank correlation coefficient between the observed intensities of thermal and epithermal neutrons could be derived when several study sites are investigated. Here, it would be highly important that the detectors are shielded consistently (e.g. a polyethylene shielding only for the epithermal counter tube) and that the neutron intensities are always corrected in the same way for e.g. atmospheric pressure variations. A very important point is that the correlation between thermal and epithermal neutrons contains

20   information on potential differences between the near-field and the far-field of the neutron detector but it remains unknown if this is a suitable approach when the soil moisture patterns are distributed differently. For instance, if the neutron detector is placed on the border between two soil moisture patterns with moist soils covering one half and dry soil covering the other half of the footprint, the relationship between the observed intensities in the two energy ranges is likely to be different. Thus, more research is certainly required!

25

**R4-2:** L55: The authors rightly point out here that there are methods using the ratio of epithermal and thermal neutrons to estimate biomass in the sensor footprint (e.g. Tian et al., 2016). The site description seems to suggest that there is a uniform (spatial) biomass at the test site, it would be better to explicitly state this if true. The literature has shown the ratio of thermal and epithermal neutrons can be influenced by biomass changes (although research tends to be looking at this temporally rather

30   than spatially), so knowing that biomass is spatially uniform at the site would be beneficial. On this point I feel a bit more discussion on possible impacts of spatially diverse biomass would make the paper more robust, considering research has shown

biomass to impact neutron ratios too. The limitations are touched upon (L586) but an expansion on this (hypothesis of influence, future research ideas?) would benefit the paper.

**A4-2:** This is correct. As it can be seen on the fig. 1, the very most fraction of the footprint is permanently covered with grassland. As we do not have information on the amount of biomass stored above ground, we assume it to be constant in time. This is reasonable as on permanent grassland no large agricultural management practices, except cattle grazing, and no large biomass changes due to cropping and harvest occurred.

We agree with the reviewer that the relationship between thermal and epithermal neutrons is likely to be influenced by distinct differences in vegetation patterns and dynamics. This influence then depends on the soil moisture contents and will be larger at observation sites with dry soils. Due to the moist conditions at our study site and most parts being covered with grassland, we expect no large influences for the present study. However, in future studies investigating the relationship between epithermal and thermal neutrons in greater detail using field data and neutron simulations should include the influence of hydrogen stored in vegetation. This would benefit both, deriving soil moisture and biomass from CRNS. We will add some information at appropriate locations in the manuscript.

**R4-3:** L141: Three simplifications in the model are outlined here. A brief expansion on the impact the authors predict this may have on the simulation would be a benefit to the reader.

**A4-3:** This was already mentioned by a previous reviewer regarding the definition of the detector in the model. We will add information for clarification.

**R4-4:** L60: Needs re-wording as it sounds a bit confusing currently: perhaps something like "However, the integration radius of thermal neutrons at the CRNS sensor can be expected to be much smaller (a footprint of approx. 35m)"

**A4-4:** We agree. We thank the reviewer for this suggestion and will replace the existing statement.

Original: "However, the average footprint size of CRNS, e.g., the integration radius of thermal neutrons can be expected to be much smaller (approx. 35 m) compared to epithermal neutrons (200 m) (e.g. see, Bogena et al., 2020)."

Adjustment: "However, the integration radius of thermal neutrons at the CRNS sensor can be expected to be much smaller (a footprint of approx. 35m) compared to epithermal neutrons (200 m) (e.g. see, Bogena et al., 2020)."

**R4-5:** L109: Write the actual value for the material density of quartz next to the description.

**A4-5:** We will add the density of quartz used for calculating the porosities from soil samples taken in the field.

---

## Author Response (AR1)

**Revision of the manuscript "hess-2021-202"**

This document contains a point-by-point reply to the comments of all reviewers, corresponding author responses and adjustments made in the original manuscript.

- 5 The major comments of the four anonymous reviewers concerned the neutron transport simulations as well as the combination of thermal and epithermal neutrons for an improved estimation of near-field soil moisture using CRNS at our study site. We added and modified paragraphs throughout the manuscript in order to clarify aspects to the reader and improve the overall manuscript.
- 10 The reviewer comments are denoted with R1-1 (reviewer 1, comment 1) and A1-1 (author response to reviewer 1, comment 1). When changes are made to phrases in the original manuscript, we state the original and adjusted sentence/paragraph or provide the phrase added.

**Reviewer #1:**

We thank reviewer #1 for taking the time to review our manuscript and their valuable suggestions, ideas and positive comments regarding our manuscript. We especially liked the idea of large-scale sprinkling experiments as it is already planned to investigate the response of neutron intensities in different irrigation experiments in future studies. We are certain that these comments and corresponding adjustments greatly improve our manuscript.

R1-1: The figure legends and axis labels are too small. Please enlarge before publication.

A1-1: We enlarged the labels in all figures.

R1-2: L444. The authors argue that the thermal neutron footprint may be significantly deeper than the epithermal range of ~30-40 cm. If this is indeed the case additional profile sampling of soil chemistry is needed by the community in order to understand the distribution of trace elements (e.g. Gd and B) that may greatly impact the thermal neutrons. In particular, as the neutrons interact with more soil horizons (beyond the O and A typical for epithermal) soil chemistry may play a greater role. The authors point this out a little but should highlight the need by the community to sample more soil horizons for relevant epithermal and thermal neutron soil chemistry. Avery et al. 2016 and others have presented a nice lattice water dataset for the top 30 cm but it seems the community needs to expand this effort across CRNS sites and more soil horizons.

A1-2: The integration depth of thermal neutrons remains under debate. We found that the average measurement depth ( $D_{86}$ ) strongly depends upon the definition of integration depth. If the point of thermalization (i.e. the point where a neutron first reaches a thermal energy) is used, the measurement depth is slightly shallower then the one of epithermal neutrons while it becomes much larger when the maximum depth along the complete neutron transport path is used. We intended to show that more research is needed in order to understand the thermal neutron transport under different environmental settings and cannot deliver a final solution on the definition of the integration depth of thermal neutrons. We added the following sentence to the revised version of this manuscript.

Original: "Consequently, thermal neutrons might contain information of soil moisture from even greater depths than epithermal neutrons (see fig. 3 and 6). However, little is known about the vertical and horizontal footprint dynamics of thermal neutrons in general and care should be taken when interpreting the presented results based on a heterogeneous model domain and a narrow range of simulated boundary conditions."

Adjustment: "Consequently, thermal neutrons might contain information of soil moisture from even greater depths than epithermal neutrons (see fig. 3 and 6). This would have implications on e.g. soil sampling campaigns for calibration as larger sampling depths might be required if thermal neutrons are of interest. However, little is known about the vertical and horizontal footprint dynamics of thermal neutrons in general and care should be taken when interpreting the presented results based on a heterogeneous model domain and a narrow range of simulated boundary conditions."

R1-3: L475. The authors show the heteroscedasticity effect from local and far-field soil moisture changes on the thermal and epithermal scatterplots nicely in Fig 8A. Without the soil moisture data in the peatlands, the conclusion is somewhat more
speculative based on GW depth but still compelling. However, additional CRNS sites with largescale irrigation (60 ha) from center pivots may confirm this effect (CRNS sites exist in NE, KS, and IA in the USA with center pivots). As the center pivots water in pie slices over 48-72 hours they will create this near and far-field effect, particularly when compared against a rainfall event on-site. The authors could mention this experiment as future work needed by the CRNS community to help confirm the conclusions here.

**A1-3:** We agree with the reviewer, and had in fact already included a similar idea in a proposal. Accompanied with intensive soil sampling or in-situ sensor networks as well as neutron transport simulations, such experiments could also help in finding supporting evidence for the measurement radius of thermal and epithermal derived from neutron transport simulations in numerous studies during the past decade. For instance, and with respect to this study, such an experimental setup could help finding an answer of whether using the point of thermalization, the point of the first soil contact or a different characteristic of a simulated detected thermal neutron for calculating the footprint radius. We added this information at the end of chapter 4.2 (L. 496) of a revised version of this manuscript as stated below.

Original: "However, this requires further research regarding the relationship of thermal and epithermal neutron intensities under changing soil moisture and with respect to different environmental factors such as soil chemistry. This may also require the need to develop transfer functions for thermal neutrons similar to those already available for estimating soil moisture from epithermal neutron intensities (e.g., Desilets et al., 2010; Franz et al., 2013; Köhli et al., 2021)."

Adjustment: "However, this requires further research regarding the relationship of thermal and epithermal neutron intensities under changing soil moisture and with respect to different environmental factors such as soil chemistry. As study sites are always restricted to local boundary conditions, large scale irrigation experiments using e.g. center pivots (see also Franz et al. 2015) combined with neutron transport simulations could improve and extend the insights gained in this study regarding indicators for footprint heterogeneity as well as the definition of measurement footprints in general. Additionally, this may also require the need to develop transfer functions for thermal neutrons similar to those already available for estimating soil moisture from epithermal neutron intensities (e.g., Desilet set al., 2010; Franz et al., 2013; Köhli et al., 2021)."

**R1-4:** Need space "time seriesNET"

A1-4: We added a space.

R1-5: Figure 3. check legend and data time series in panel B? No red squares etc.

A1-5: In figure 3, panel b), the colours are correct but indeed the point symbols are not. We apologize for this mistake and corrected the symbology.

**R1-6:** L538. Also, watch out for overfitting if only calibration data is available on campaign days. The CRNS community has found ~3 calibration dates are needed for robust N0 estimation. For 4 parameters you may need 12 or more calibration sampling days. That is a lot of digging :). It is already challenging to calibrate on multiple days if you have several CRNS sites.

A1-6: We agree with the reviewer. As stated in the manuscript (L. 540), adjusting the parameters requires a sufficient spatiotemporal coverage of a sufficient number of reference measurements. In this study, permanently installed point sensors were used which provide continuous data and thereby support the calibration process. We believe that further soil sampling campaigns would be time-consuming, labour-intensive, and would not add much additional information to the already continuous dataset.

**Reviewer #2:**

We thank reviewer #2 for taking the time to review our manuscript and their valuable suggestions, ideas and positive comments regarding our manuscript. We are glad about the comments regarding the combination of both neutron energies which lead us to add further information for clarification. We are certain that these comments and corresponding adjustments greatly improve our manuscript.

R2-1: "The more complex response of the thermal neutrons rise my main concern of the present study. The potential use of
thermal and epithermal neutrons has been pointed since the first CRNS publications (Desilets et al., 2010; Rivera Villarreyes et al., 2011). Further attempts have also been performed later by dedicated studies (Bogena et al., 2020; Jakobi et al., 2018; Tian et al., 2016). As far as I understood, difficulties to handle these two signals are related to the non-unique response of the thermal neutrons, i.e., in contrast to epithermal, they depend on chemistry and the thermal intensity also increases during the wetting of initially dry soils (Desilets et al., 2010; Zweck et al., 2013). The present study quantifies the different footprint of
thermal and epithermal neutrons. As such, it sheds lights on the understanding on the processes. However, few is discussed on the possibility to generalize the correlation found in the present study between epithermal and thermal in other conditions. As such I see the high risk of this study to be very limited. In addition, if the Authors are really interested on disentangling footprint variability, I rather believe that the use of side-shielded detector (Zreda et al., 2021) could be easier and more promising than the use of the thermal detector."

A2-1: As the reviewer pointed out and as we stated in the manuscript, the response of thermal neutrons to soil water changes is more complex and does also depend on soil chemistry. As pointed out by Weimar et al. (2020), apart from the moderation optimum there is a general decrease of the thermal neutron intensity with increasing soil moisture but with a less steep slope compared to epithermal neutrons. At the study site described in the present study, the neutron response of epithermal neutrons

- 25 is too steep for the soil moisture changes observed in mineral soils close to the neutron detector. For this reason and at our study site, we can make use of thermal neutrons as a proxy for a different relationship between neutron intensity and soil water changes. In this case we agree with the reviewer, that the methodological approach for combining both neutron energies through the rescaling approach presented, is limited to the conditions of the study site and study sites with a similar setting. For this reason, we also successfully tested and presented another approach for improving the calibration against near-field
- 30 reference measurements which does not involve thermal neutrons and is generally applicable. In contrast, using the relationship between thermal and epithermal neutrons for the identification of footprint heterogeneity between the near-field and the far-field of the neutron detector is not limited by the more complex behaviour and non-unique response of thermal neutron to soil water changes because at study sites with homogeneous soil water states and dynamics, the relationship should not change. On the contrary, if soil moisture contents and patterns differ, differences should become visible as illustrated in Figure 8. This approach as a first indicator for footprint heterogeneity in terms of differences between the nearfield and far-field of the instrument and could be applied at other study sites. Here, we would like to refer to the comment of reviewer #1 (R1-3) suggesting centre pivot irrigation experiments to test this indicator under more controlled conditions. Directional neutron detectors pose a great potential to the scientific community as they allow for reducing the effect of the soil,

- 5 when a thick moderator shield is placed below the detector or intensify the soil moisture signal when the moderator shield is placed above the detector. Side-looking devices need to be tested in future studies. A potential drawback could be the fact that neutrons scatter and change directions several times before actually entering the detector. As a consequence, for example, a neutron being detected by a north-ward looking detector does not necessarily originate from this direction or had most of its elastic scattering interactions in this direction. This is different for a directional downward-looking neutron detector placed
- 10 above the soil. Here, most neutrons are directly reflected from the soil and thus carry information of soil water contents. Nevertheless, the potential of directional neutron detectors for side-looking applications as well as CRNS in general should be explored in future. Research progress develops rapidly and during the time of this review, new research has been published. We added a phrase in the discussion section mentioning the work of Badiee et al. (2021) and Francke et al. (2021) regarding directional neutron detectors.
- 15

Original: "Hence, the second alternative approach tested in this study is suitable for a separation of near and far field soil moisture at the heterogeneous observation site investigated.

Several limitations need to be considered when assessing the improvement achieved with the different approaches tested in this study."

Adjustment: "Hence, the second alternative approach tested in this study is suitable for a separation of near and far field soil moisture at the heterogeneous observation site investigated.

"Both approaches tested in this study allow for an improvement of the estimation of near-field soil moisture and illustrate the potential for separating the measurement footprint where approach 1 is generally applicable while approach 2 may be most suitable at sites with conditions similar to those at the study site investigated. We would like to note that further instrumental adjustments based on additional shielding may also offer the potential for limiting the measurement footprint. These include downward-looking detectors (Badiee et al. 2021) or side-looking devices (Francke et al. 2021). However, using a non-modified detector conserves the possibility to also retrieve an area-average soil moisture time series in the entire footprint.

Several limitations need to be considered when assessing the improvement achieved with the different approaches tested in this study."

**R2-2:** As far as I have understood, the comparison between neutrons simulations and neutron measurements is not consistent. Simulations are based on theoretical detectors sensitive only to thermal or epithermal ranges. In contrast, measurements have been collected with bare and moderated detectors that are contaminated by epithermal and thermal neutrons, respectively, as highlighted by the Authors. Previous studies showed clear discrepancy between simulations and measurements when this contamination effect was not properly account for (Andreasen et al., 2016; McJannet et al., 2014). As such, I'm surprised about this setting. Either the detectors should be improved to remove the contamination from thermal and epithermal. If this is not possible within the present study, why not repeating the simulations with the real detectors? Despite more rigorous understanding of thermal and epithermal, you are allowed to compare the simulations and the experimental data.

A2-2: We partly agree with the reviewer that the contribution of epithermal neutrons to the thermal neutron detector and vice versa influence the observed neutron intensities which hampers a direct comparison between observed neutron intensities and

- 10 those obtained from neutron transport simulations. An option could be the use of detector response functions which mimic the sensitivity of a real neutron detector. However, we are more interested in accounting for the actual thermal and epithermal neutron signals at the detector location in the simplified scenario for deriving a general understanding of the expected neutron intensities under heterogeneous site conditions than in reproducing the signal that our real-world sensor would have measured. As a consequence, an ideal virtual neutron detector is more useful for our objective. Furthermore, due to missing soil moisture
- 15 information from peatland areas as well as a number of simplifications made in the model which, for instance, include the identical chemical composition of organic peatland soils and mineral soils it is not possible to generate a reasonable fit between simulation and observation. For example, the additional hydrogen and carbon stored in peatland soil need to be considered when trying to simulate real-world data and was not possible with the model version used. Similarly, missing trace elements such as gadolinium and boron in mineral soils will have an influence on the simulated and observed neutron intensities in the
- 20 thermal domain. Due to these model simplifications we decided to use the energy window option in order to understand the neutron flux at the detector location in the thermal and epithermal energy range and draw more general conclusions for such heterogeneous soil water distributions. We added a paragraph to chapter 2.2.1 of the revised manuscript illustrating this reasoning.
- 25 Added phrase: "Furthermore, in this study, we consider an energy window for defining thermal and epithermal neutrons scattering in the virtual detector. Although detector response functions (e.g. see Köhli et al. 2018, 2021) mimic the sensitivity of a real neutron detector and would provide a more realistic neutron intensity, we deliberately decided to keep our model simple and general (also due to lack of information on soil moisture dynamics and soil chemistry in the peatlands). We are thus not aiming at a detailed reproduction of field conditions but at more general understanding."

**R2-3:** Point scale soil moisture observations are very limited and they represent only short distance. This has been pointed as main limitation of the present study but very late in the manuscript and without explaining the consequences of that. Please note that the use of limited number of soil moisture locations have been highly criticized in former studies (see discussion for (Rivera Villarreyes et al., 2013). Despite I'm personally do not against comparisons with relative few points sensors, it should be noted that the present study concluded that the use of thermal signal improved the performance. However, I see a strong bias if we consider that thermal has a smaller footprint and point scale soil moisture used for the comparisons are located in the near field. As such, I would rather presume to have worst results in case the point soil moisture sensors would have been distributed also at larger distance.

A2-3: The number of permanently installed in-situ references soil moisture sensors is limited and an assumption we have to make is that the average soil water content observed by the limited number of reference sensors is valid for the part of the footprint covered with mineral soils. A supporting indicator for this assumption is that the two different sensor types installed

- 10 show similar absolute average values and soil water dynamics. Deriving an aerial average soil moisture from calibration against several in-situ sensors distributed throughout the footprint was not possible due to the low number of sensors. Thus, the aim of this study was to improve the calibration against reference sensors in the near-field in order to make a step forward in deriving a spatially differentiated soil moisture time series from CRNS. Therefore, we conducted the neutron transport simulations in order to understand the footprint sizes of thermal and epithermal neutrons as well as the influence of soil 15 moisture variations in the simulated neutron intensities of both energy ranges.
- The smaller footprint of thermal neutrons as well as the weaker increase of thermal neutrons with decreasing soil moisture can be seen in the model results. Thus, thermal neutrons can be used as a proxy to produce a rescaled neutron count rate which improves the calibration. The applicability of this method may be indeed site-specific and may not produce improved results at other study sites with different soil water distributions and environmental settings. Therefore, we also tested the approach
- 20 of adjusting the parameters of the standard transfer function as a more transferrable method. Although it might be site-specific, the combination of thermal and epithermal neutrons presented here can illustrate the potential different neutron energies and of having a closer look on thermal neutrons for the estimation of soil water contents in the scope of CRNS.
- 25 R2-4: L2: I suggest the term estimations instead of measurements, i.e., the sensors measures neutrons and, based on that, estimate soil moisture.
  - A2-4: We agree. We changed "measurements" to "estimations" in line 2.
- 30

R2-5: I think it should better phrased. 1) soil homogeneous conditions are unlikely and from my understanding 2) the added value of CRNS emerges exactly in case of heterogeneous conditions. The key assumption is in my opinion to sense a representative volume where soil moisture shows a relative short correlation length. In this case neutrons well mix within the footprint. In case of longer correlation length and spatial patterns, empirical data deviates from theoretical functions and hysteresis behaviour could also emerge. Similar consideration has been detected in snow patches conditions (Schattan et al., 2019)

A2-5: We agree with the reviewer in saying that soil homogeneous conditions are unlikely, and we did not claim this in the manuscript. Nevertheless, there is no approach for translating observed neutron intensities into soil moisture yet that explicitly considers sub-scale heterogeneity in the CRNS footprint. Usually it is assumed that soil moisture is homogenous or that its spatial correlation length is smaller than the CRNS footprint. However, given the non-linear relationship between neutron counts and soil moisture, this assumption my not be applicable for calculating CRSN footprint soil moisture. We agree that this should be clarified in the revised manuscript and made the following adjustment.

Original: "Most approaches and processing techniques for observed neutron intensities are based on the assumption of homogeneous site conditions within the measurement footprint of the neutron detector."

Adjustment: "Most approaches and processing techniques for observed neutron intensities are based on the assumption of homogeneous site conditions, or of soil moisture patterns with correlation lengths shorter than the measurement footprint of the neutron detector. However, in view of the non-linear relationship between neutron intensities and soil moisture it is questionable whether these assumptions are applicable."

**R2-6:** L39. If I'm not wrong, some papers refer to the threshold 0.5 for thermal neutrons. Could you provide reasoning for this value?

A2-6: The threshold of 0.5 eV in some previous studies refers to the cutoff-threshold for cadmium as an additional shielding material to reduce the contribution of thermal neutrons to the moderated (epithermal) counter tube of the neutron detector. In

- 25 this study we did not use a second shielding (cadmium or gadolinium) in addition to the polyethylene shielding of the moderated (epithermal) counter. Due to the absence of this shielding in neutron observations made on-site, we decided to use the physical energy threshold of thermal neutrons in the neutron simulations; i.e. the energy were thermal neutrons are going to be in equilibrium with the surrounding atomic nuclei energetically and neutron absorption (capture) becomes a relevant process. For further clarification we added the following phrase to chapter 2.2.1 the revised manuscript.
- 30

Added phrase: "Although different studies refer to different upper energy boundaries for thermal neutrons in simulation studies which depend on the specific neutron detector and applied shielding material, we decided to use the physical energy threshold in which thermal neutrons are in equilibrium with the energy of environmental nuclei and neutron absorption becomes a relevant process as a more general definition of thermal neutrons."

**R2-7:** L43. The more complex response of the thermal neutrons rise my main concern of the present study, i.e., the results are very site specific (see general comment above)

**A2-7:** The more different response of both energy ranges (see also chapter 1 and 4.1) needs to be indeed considered and requires further and more general investigations to enable further applications of thermal neutrons in CRNS. A combination of thermal and epithermal neutrons leads to an improved estimation of near-field soil moisture at our study site (see our responses above). Although the rescaling approach presented might be rather situation-specific and the combination of thermal

- 10 and epithermal neutrons presented in this study may not lead to improvements at study sites with different soil moisture conditions and spatial patterns, the second alternative calibration approach does allow improving the calibration against nearfield reference measurements without the use of thermal neutrons. Thus, our study is able to present a more general approach for the improved calibration of near-field soil moisture as well as illustrating the potential of thermal neutrons which should be explored in greater detail in future studies. Here, it would be inevitable to consider and investigate the impact of the non-
- 15 unique behaviour of thermal neutrons in order to estimate more general implications for the use of thermal neutrons in the scope of CRNS. Although this is beyond the scope of the present field study, further research investigating the thermal neutron characteristics in greater detail are underway.
- 20 **R2-8:** L65. I'm very surprised if most studies with stationary CRNS assume homogeneous site conditions. Please rephrased as previously discussed.

A2-8: As done in A2-5, we rephrased the sentence.

Original: "This may be of particular importance as most studies with stationary CRNS assume homogeneous site conditions."

Adjustment: "This may be of particular importance as most studies with stationary CRNS assume quasi-homogeneous site conditions or spatial patterns of different soil moisture states and dynamics with correlation lengths smaller than the CRNS footprint."

R2-9: L105. Please add if possible the 21 random locations on figure 1.

A2-9: We added the sampling point locations to figure 1.

R2-10: L125. but why to simulate something that it does not represent the real detector? See general comment above

A2-10: Please see to our detailed response A2-2 and the adjustments made.

**R2-11:** L131. point scale soil moisture observations are very limited, they represent only short distance and they are not evenly distributed. The CRNS calibration is strongly biased

A2-11: The reference sensors being installed close to the sensor only as well as the missing reference sensors in far-field peatland soils are indeed important limitations of this study. However, because both sensor types installed show similar dynamics we assume that they are representative for the area covered with mineral soils. The limitation of reference soil moisture sensors that are limited to the mineral soil part is discussed in chapter 4.3.

R2-12: L166-183. This text refers to all simulations and not only to simulation set 1. It should be moved up in section 2.2.

A2-12: We agree. We moved the lines 166-183 in the original manuscript to section 2.2 starting after line 149. Note that added sentences in this part of the manuscript which are based on previous comments are moved to the new location as well.

**R2-13:** L180. As far I understood from previous studies, D86 is not spatially constant. Please specify if you refers here to the maximum (or average) depth over the footprint

A2-13: This is correct. The integration depth decreases with increasing distance to the neutron detector. In our study  $D_{86}$  represents the average depth in the footprint per simulation scenario. We rephrased this in the following way:

Original: "For thermal neutrons, the measurement depth  $D_{86}$  is defined as the 86 percent quantile of either the depth of the thermalization point or the maximum depth along the neutron transport path while for epithermal neutrons we use only the latter."

Adjustment: "For thermal neutrons, the average measurement depth  $D_{86}$  is defined as the 86 percent quantile of either the depth of the thermalization point or the maximum depth along the neutron transport path while for epithermal neutrons we use only the latter."

**5**

**R2-14:** Figure 2. ground water level should be reported as depth from soil surface to ground water instead of ground water level above sea level to facilitate the interpretation on the discussion on shallow water table influencing soil moisture detected by CRNS.

A2-14: The groundwater levels displayed refer to the height above sea level. However, peatland areas have different elevations (see figure 1) ranging between e.g. between 59 and 61 meters and mineral soils areas being located slightly higher. In all concerning figures, we replaced the groundwater level with the groundwater table depth. We decided to use an estimated average elevation of peatland areas in the footprint of 61.5 m to calculate the approximate groundwater table depth.

**R2-15:** Equation 6: you merge thermal and epithermal with different footprints. But you compare the scaled sum with point scale soil moisture weighted based on epithermal footprint. Are you not mixing up the signals? Additionally, what about using only thermal? I expect good or even better results when calibrating with these near field point locations.

- 20 A2-15: This is correct. We intended to mix the signals to derive a new rescaled signal. This signal is then more similar to the theoretical epithermal intensity occurring if the entire footprint would have the soil moisture conditions of the near-field of the sensor and thus, better matching the shape of the functional relationship of the standard transfer function. One reason is the smaller footprint of thermal neutrons making them less influenced by far-field soil moisture variations. Another more important reason for the improvement is the generally smaller decrease of thermal neutrons with increasing soil moisture. As a
- 25 consequence, we can make use of thermal neutrons as proxy for a different signal response better matching the response which would occur if the soil moisture conditions of the near-field would cover the entire footprint at our site. As described earlier, this approach may not be directly transferable to sites with different spatial patterns of soil moisture and different dynamics. However, the more general approach by adjusting the transfer function instead of adjusting the neutron signal could be used instead.
- 30 We did not consider thermal neutrons alone as this would require to derive a new transfer function. This, however, requires extensive modelling efforts and is beyond the scope of this study. Nevertheless, this should be considered in future studies; especially when thermal neutrons applications evolve and the observed thermal neutron intensities need to be explained in detail. This would then require a closer look on soil chemistry as well as it was pointed out by reviewer #1 and #2. We added a sentence to section 2.3.2.

Added phrase: "We would like to note that in this study, we did not consider using thermal neutrons alone as this would require a transfer function specifically designed to the response of thermal neutrons to changes in soil moisture."

**R2-16:** Figure 3b. If you calculate the maximum D86, it should be expected to not changing much the depth by increasing far-field soil moisture. Please clarify

A2-16: As clarified in author response A2-13 the measurement depth D86 represents the areal average measurement depth. Under dryer conditions in peatland soils, the footprint radius is larger and a stronger influence of peatland soils on D86 can be expected. For this reason, the measurement depth remains stable if the near-field soil moisture remains stable and the far-field soil moisture is high. However, when soil moisture in far-field areas is generally low, the influence of far-field areas is stronger and small changes in the average measurement depth are visible. Please also see the adjustments made based on A2-13.

**Reviewer #3**

We thank reviewer #3 for taking the time to review our manuscript and their valuable suggestions, ideas and positive comments regarding our manuscript. We are glad about the comments regarding the combination of both neutron energies as they were

- 5 similarly stated by reviewer #2. Furthermore, the proposed equation in addition to the approach proposed by Schrön et al. (2018b) pose an interesting starting point for further research concerning the disaggregation of the measurement footprint. We are certain that these comments and corresponding adjustments greatly improve our manuscript.
- 10 **R3-1:** Fig 6a: Why is there so little change in the R86 footprints in this figure (as measured by first soil contact) compared to Fig 3a? In the case of simulation set 3 (dashed lines Fig 6a) far field soil moisture drops from 0.7 to 0.2 which is very similar to the drop from 0.7 to 0.1 in Fig 3, but in Fig 6 we also have that the near field soil moisture is decreasing. Surely there should therefore be a larger change in the various footprints in Fig 6? In fact R86 actually decreases slightly in simulation set 2 between scenario 1 (wettest) and scenario 6 (driest)! Is there a mistake? Have missed something important?

A3-1: Thank you for this comment! We checked the postprocessing procedure of the simulation data of all simulation scenarios conducted and found a mistake in the code related to the processing of the simulated neutrons (i.e. which neutrons are considered as detected). Thus, not all intended neutrons were counted leading to higher total intensities in the simulations and different footprint sizes. Furthermore, we did not use the border of the virtual detector to calculate the distance to the point of origin of a neutron but the detector centre. We also changed this as this leads to more intuitive radial distances. Fixing the code leads to different absolute simulated neutron intensities and radial footprints. For instance, the average footprint radius of

- epithermal neutrons in simulation set 1 decreases from 134 to 121 m as we now also use the detector boundary instead of the detector centre for calculating the distances to the point of origin of each neutron. Although the absolute values are different, in relative terms, the results are very similar to those already in the manuscript and thus, fixing the code does not lead to a
- 25 different interpretation of the simulation results. This also confirms the low sensitivity of our results to the site-specific footprint size. Nevertheless, we apologize for this mistake and will update all values throughout the manuscript and update all figures. The updated figures are shown below.

In respect to Figure 6, correcting the code does not have a major effect. First of all, simulating a relatively small virtual detector in a large model domain leads to a limited number of simulated neutrons actually reaching the neutron detector compared to e.g. measuring the total number of neutrons in a certain energy range are reflected from the soil in the entire model domain which cannot be done under heterogeneous soil moisture conditions. Consequently, a certain degree of statistical variations has to be considered. This is also displayed in Figure 4 and can explain certain variations visible also in the figures showing the calculated  $R_{86}$  and  $D_{86}$ . This is likely to be the most important reason for the slight decrease and variations visible for the  $R_{86}$  in fig. 6a for epithermal neutrons.

The footprint change of epithermal neutrons differs between the simulation set 1 and those of simulation set 2 and 3. This can also be related to the fact of the footprint change becomes smaller with more water (i.e. hydrogen) being in the model domain.

Due to this non-linearity, the largest footprint changes can be expected under rather dry conditions from e.g. 0-0.15 m3 m-3. As a consequence, one reason for the small differences in footprint changes can be related to the high soil moisture contents simulated and adds to the statistical uncertainty of the Monte-Carlo simulations mentioned before which has a higher impact if the footprint change with changing soil moisture is smaller.

When more water is located close to the detector, where it is most sensitive, fewer neutrons reach the detector and the statistical

- 10 noise increases. However, simulating a discrete virtual neutron detector requires a very high number of source neutrons to be simulated and is computationally intensive. This study is a first investigation of the influence of soil moisture patterns on the radial footprint sizes and thus, a full and general analysis of the measurement footprint of thermal and epithermal neutrons is beyond the scope of this study, but underway for thermal neutrons in a very recent publication of some of the authors (Jakobi et al. 2021). Despite the limitations regarding the accuracy of the simulation results compared to the real-world site, they still
- 15 allow valuable conclusions within the scope of this study, that e.g. thermal neutrons have a smaller footprint radius, that farfield soil moisture variations still have an influence on the thermal neutron count rate and thus, that the definition of the origin for calculating the footprint radius (e.g. point of thermalization or point of first soil contact) needs to be investigated further – especially under heterogeneous distributions of soil water in the model domain. Lastly, the influence of the geometry and spatial distribution of hydrogen (-variations) in respect to the detector location remains largely unknown and could also be a
- 20 reason for the partly inconclusive behaviour of calculated radial measurement footprints for the simulations with an equal decrease of soil moisture. The anisotropy of CRNS footprints was already described in Schattan et al. (2019). We updated all concerning figures as well as stated values e.g. for footprint sizes and intensities throughout the manuscript.

**Updated figures:**

Figure 3. Simulation results for the measurement footprint radius (a) and depth (b) of detected thermal and epithermal neutrons.

**Figure 4**. Total number of neutrons in the thermal and epithermal energy range observed by the virtual detector per simulated peatland soil moisture.

---

## Author Response (AR2)

**Revision (2nd iteration) of the manuscript "hess-2021-202"**

This document contains a point-by-point reply to the comments of all reviewers, corresponding author responses and adjustments made in the second iteration of the revised manuscript.

5   The reviewer comments are denoted with R1-1 (reviewer 1, comment 1) and A1-1 (author response to reviewer 1, comment 1). When changes are made to phrases in the original manuscript, we state the original and adjusted sentence/paragraph or provide the phrase added.

We thank reviewer #2 for taking the time to review our revised manuscript and their additional valuable suggestions, ideas and positive comments regarding our manuscript.

**R2 (General comment):** The Authors addressed all my comments and the manuscript has been further improved. Despite I still believe that a more consistent approach should be considered to compare simulations and measurements, in my opinion the present manuscript is suitable for publication.

I encourage the Authors to overcome this limitation in future studies to shed new lights on the topic, e.g., simulations should consider real heterogeneous soil moisture conditions, detector response function etc.

**A2 (General comment):** Thank you very much for your comment! In future studies addressing the heterogeneous soil moisture distributions on the CRNS response, we will include more realistic soil moisture distributions based on additional field observations in neutron transport simulations. In this case, considering detector response functions is highly necessary in order to better reconstruct the observed neutron signal and will be incorporated in the analyses. However, in the present study we would like to show more general results allowing to draw more general conclusions for sites with a heterogeneous (binary) soil moisture distribution instead of reproducing the neutron intensity observed at a specific study site.

**R2-1:** L255: how was lattice water measured? I would add this information at L115

**A2-1:** We added this information in the following way:

Addition: "A subsequent loss-on-ignition analysis (1000 °C, 24 hours) revealed an average content of lattice water of 0.03 g g$^{-1}$ for organic and 0.001 g g$^{-1}$ for mineral soils, respectively."

**R2-2:** L271: what do you mean exactly by quasi-random?

**A2-2:** We use the term quasi-random to express that the random values generated in algorithms are not fully random due to the nature of the random number generating algorithm. For instance, it allows to set seeds for reproducibility. However, for simplicity, we changed we term to "random" in the entire manuscript.

**R2-3:** Figure 9 shows same as figure 8a with in addition the color points. Thus Figure 9 could be merged with fig.8.

**A2-3:** This is correct. However, as we have stated in the point-to-point reply of the initial revised manuscript, we would like to keep Figure 9 to a separate one for better visibility of the figure's content.

**R2-4:** L466: "After" lower case.

**A2-4:** Done.

**R2-5:** L526: two times "and". To remove one

**A2-5:** Done.

**R2-6:** L559: if you want to enrich with additional uncertainty assessments you might also add: . Baroni, G., L. M. Scheiffele, M. Schrön, J. Ingwersen, and S. E. Oswald. "Uncertainty, Sensitivity and Improvements in Soil Moisture Estimation with Cosmic-Ray Neutron Sensing." Journal of Hydrology 564 (September 1, 2018): 873–87. https://doi.org/10.1016/j.jhydrol.2018.07.053.
. Schattan, P., M. Köhli, M. Schrön, G. Baroni, and S. E. Oswald. "Sensing Area-Average Snow Water Equivalent with Cosmic-Ray Neutrons: The Influence of Fractional Snow Cover." Water Resources Research 55, no. 12 (December 2019): 10796–812.https://doi.org/10.1029/2019WR025647.
. Andreasen, M., K. H. Jensen, D. Desilets, M. Zreda, H. R. Bogena, and M. C. Looms. "Cosmic-Ray Neutron Transport at a Forest Field Site: The Sensitivity to Various Environmental Conditions with Focus on Biomass and Canopy Interception." Hydrol. Earth Syst. Sci. 21, no. 4 (April 3, 2017): 1875–94. https://doi.org/10.5194/hess-21-1875-2017.

**A2-6:** We agree that these publications should be mentioned in line 559 and added them accordingly.

**R2-7:** L577: point instead of colon?

**A2-7:** We replaced the colon.

**R2-8:** L618: two times "the". To remove one

**A2-8:** Done.

**Reviewer #3:**

We also thank reviewer #3 for taking the time to again review our revised manuscript and their additional valuable suggestions, ideas and positive comments regarding our manuscript.

**R3 (General Comment):** I thank the authors for their replies and actions on my comments, especially the checking on Figure 6 (comment R3-1) which, for me, was my only real concern. Actually, the source of my confusion about this figure was my assumption that greater soil moisture anywhere in the simulation domain would always reduce the epithermal footprint. Actually this appears not to be the case if the soil moisture is increased in the near field! This was a novel point to me. And perhaps the authors would like to comment on it. To see this compare epithermal R86 footprints of simulation 1 in set 1 (111 m) and that from set 3 (128 m). I think the lack of change in the epithermal R86 could then be explained by a competition between increasing far-field soil moisture which would reduce R86 and increasing near field soil moisture which would increase it?

**A3 (General Comment):** Thank you for your comment. Yes, an increasing soil moisture reduces the neutron count rate and thus, the footprint size in the epithermal energy range. This effect is also influenced by the interplay of the different absolute states of soil moisture and their distribution in the model domain.

In Figure 6, we show the results of the simulation sets 2 and 3, where we decrease the soil moisture in both, the near-field and the far-field but at different rates (solid and dashed blue lines). In the near-field, we reduce the soil moisture from 0.35 to 0.1 $m^3$ $m^{-3}$ in both simulation sets. However, in simulation set 2 (solid line), the far-field soil moisture is reduced to a minimum value of 0.45 and for simulation set 3 (dashed line) to a minimum of 0.2 $m^3$ $m^{-3}$.

Consequently, in absolute terms, the minimum soil moisture scenario in simulation set 2 is much lower compared to simulation set 3. Due to the non-linearity of the response of neutron intensity and soil moisture and thus, soil moisture and $R_{86}$, we probably do not see an increase of $R_{86}$ in simulation set 2; simply due to the much higher minimum soil moisture content in the model domain.

Simulation set 2 (solid line) shows fluctuations of $R_{86}$ in Figure 6 which may be linked to the spatial distribution of absolute soil moisture contents and its variations in the model domain but could also be to some degree related to statistical noise.